# ChemRAP uncovers specific mRNA translation regulation via RNA 5′ phospho-methylation

Hélène Ipas[1,3], Ellen B Gouws[1,3], Nathan S Abell [1,4], Po-Chin Chiou [1,4], Sravan K Devanathan[1,4], Solène Hervé [1,4], Sidae Lee[1,4], Marvin Mercado[1,4], Calder Reinsborough[1,4], Levon Halabelian[2], Cheryl H Arrowsmith[2] & Blerta Xhemalçe [1✉]

## Abstract

**5′-end modifications play key roles in determining RNA fates. Phospho-methylation is a noncanonical cap occurring on either 5′-PPP or 5′-P ends. We used ChemRAP, in which affinity purification of cellular proteins with chemically synthesized modified RNAs is coupled to quantitative proteomics, to identify 5′-Pme "readers". We show that 5′-Pme is directly recognized by EPRS, the central subunit of the multisynthetase complex (MSC), through its linker domain, which has previously been involved in key noncanonical EPRS and MSC functions. We further determine that the 5′-Pme writer BCDIN3D regulates the binding of EPRS to specific mRNAs, either at coding regions rich in MSC codons, or around start codons. In the case of LRPPRC (leucine-rich pentatricopeptide repeat containing), a nuclear-encoded mitochondrial protein associated with the French Canadian Leigh syndrome, BCDIN3D deficiency abolishes binding of EPRS around its mRNA start codon, increases its translation but ultimately results in LRPPRC mislocalization. Overall, our results suggest that BCDIN3D may regulate the translation of specific mRNA via RNA-5′-Pme.**

**Keywords** RNA Modification Reader; RNA Phospho-methylation; Local Translation; BCDIN3D; LRPPRC
**Subject Categories** RNA Biology; Translation & Protein Quality

## Introduction

Although there are more than a hundred chemically distinct post-transcriptional modifications of RNAs (Boccaletto et al, 2022), their functions and contributions to human disease remain largely underexplored (Debnath and Xhemalçe, 2021), constituting one of the grand challenges of the biological and chemical fields (He, 2010). The discovery that the binding of specific proteins can be enhanced by the m6A mark on RNA (Arguello et al, 2017;

Dominissini et al, 2012; Edupuganti et al, 2017; Liu et al, 2015; Xu et al, 2014), raises the possibility that other RNA modifications can act similarly through modulating binding of specific proteins.

In addition to the internal bases, RNA modifications can also be added to the 5′- and 3′-ends of RNAs (Shelton et al, 2016). 5′-ends are particularly crucial for determining the fate of RNA molecules (Shelton et al, 2016). At the difference of the canonical m7G cap, direct O-methylation of 5′-phosphates is a chemically simpler cap occurring on the γ-phosphate (5′γ-Pme) of nascent tri-phosphorylated RNAs (7SK, U6 snRNAs), or on the α-phosphate (5′-Pme) of processed mono-phosphorylated RNAs (tRNAHis, specific precursor miRNA) (Devanathan et al, 2021). Due to the absence of available antibodies against 5′γ-Pme or 5′-Pme, the RNA targets of these noncanonical caps are not comprehensively known (Devanathan et al, 2021). 5′γ-Pme may primarily occur on nascent RNAs, with 5′ tri-phosphate ends, while 5′-Pme may primarily occur on processed RNAs, with mono-phosphate ends. tRNAHis acquires a 5′-P end after a complex molecular mechanism specific to tRNAHis, involving the consecutive action of RNase P and tRNA-histidine guanylyltransferase 1 like (THG1L) (Gu et al, 2003), while precursor miRNAs have a 5′-P as a result of RNase III Drosha-mediated cleavage of the primary microRNA (Lee et al, 2003). Other RNA processes resulting in a 5′-P end are m7G decapping (Vidya and Duchaine, 2022) and diphosphatase activities.

The 5′-Pme RNA modification is written by BCDIN3D, a member of the Bin3 family of RNA methyltransferases (Blazer et al, 2017; Martinez et al, 2017; Schapira, 2016; Xhemalce et al, 2012). BCDIN3D has been implicated in breast cancer, as its depletion significantly decreases transformation and invasion of MDA-MB-231 breast cancer cells in vitro (Xhemalce et al, 2012) and in vivo (Reinsborough et al, 2021), and its overexpression correlates with poor prognosis in breast cancer, especially in triple-negative subtypes (Liu et al, 2007; Yao et al, 2016). In addition, the BCDIN3D locus is associated with obesity and type II diabetes (Berndt et al, 2013; Reinsborough et al, 2021; Thorleifsson et al, 2009; Walley et al, 2009), suggesting a broader but unknown function in human diseases. BCDIN3D has a very high affinity for tRNAHis (Martinez et al, 2017; Reinsborough et al, 2019) and virtually all of cellular tRNAHis is 5′-Pme, even in cells with

[1]Department of Molecular Biosciences, University of Texas at Austin, 2500 Speedway, 78712 Austin, TX, USA. [2]Structural Genomics Consortium, and Princess Margaret Cancer Centre, University of Toronto, Toronto, ON M5G 2M9, Canada. [3]These authors contributed equally to this work as first authors: Hélène Ipas, Ellen B Gouws. [4]These authors contributed equally to this work as second authors: Nathan S Abell, Po-Chin Chiou, Sravan K Devanathan, Solène Hervé, Sidae Lee, Marvin Mercado, Calder Reinsborough. ✉E-mail: b.xhemalce@austin.utexas.edu

relatively low levels of BCDIN3D protein. For example, in MDA-MB-231 triple-negative breast cancer cells depleted for BCDIN3D down to 20–30% of control cells, tRNA$^{His}$ levels, 5′-Pme modification, or aminoacylation are unaffected, yet these cells display significant changes in their transcriptome, proteome, and metabolome, as well as a highly significant reduction of their tumorigenic phenotypes both in cells and in vivo (Reinsborough et al, 2021; Xhemalce et al, 2012). Because of this, it is likely that other RNA targets with fine-tunable levels of 5′-Pme are responsible for BCDIN3D depletion phenotypes.

In order to gain new insights into the mechanism of action of 5′-Pme RNA modification and BCDIN3D, we employed a high-throughput method to unbiasedly identify factors that specifically interact with 5′-Pme-modified RNA. We uncovered that 5′-Pme is directly bound by the EPRS (or EPRS1) subunit of the multisynthetase complex. Our results further suggest that this RNA modification mediates the interaction between BCDIN3D, EPRS and the multisynthetase complex to regulate the expression of specific mRNAs in a tRNA$^{His}$ independent manner.

## Results

### ChemRAP identifies 5′-Pme RNA modification readers

In order to identify proteins whose binding to RNA is modulated by the 5′-Pme RNA modification, we utilized ChemRAP, for Chemically modified RNA-Affinity Purification. ChemRAP combines the pulldown of proteins from cellular extracts with chemically synthesized RNAs to quantitative Mass Spectrometry (Fig. 1A). As noted in the introduction, we previously showed that pre-miRNAs are targets of BCDIN3D, both in cells and in vitro (Reinsborough et al, 2021; Xhemalce et al, 2012). Therefore, our pull-down probe was a synthetic chemically modified microRNA duplex, in which the 5p guide strand contained a 5′-Pme end and a biotinylated 3′-end to allow the coupling of the bait RNA to streptavidin beads (Fig. 1A). This probe is identical to a pre-miRNA on the 5′-end (Appendix Fig. S1). Two essential controls included the unmethylated microRNA duplex probe (5′-P) and the "no RNA" probe (i.e., streptavidin beads alone) (Fig. 1A,B). We employed SILAC (Stable Isotope Labeling with Amino Acids in Cell Culture) (Mann, 2006; Ong and Mann, 2006), which is particularly well-suited to maximize the sensitivity and reduce the false positive rate of methods using RNA as baits (Scheibe et al, 2012). After performing pulldowns with the 5′-P, 5′-Pme and "no" RNA baits (Fig. 1B, left panel), the miRNA-5′-P pulldown was pooled with either the "no RNA" or miRNA-5′-Pme pulldowns (Fig. 1B, right panel) in order to answer two different questions, the first interrogating for RNA binding proteins, and the second for RNA modification "reader" proteins (Fig. 1B). To eliminate the false positive hits arising from differences in the proteome of cells grown in "heavy" or "light" medium, we performed both forward and reverse experiments where each of the pulldowns was performed with either "heavy" or "light" medium. In addition, we performed these experiments with extracts derived from three biological replicates. The results of these experiments are shown on Fig. 1C,D as plots in which identified proteins are shown as a function of the log2 transformed H/L ratio in either the forward (*x* axis) or reverse (*y* axis) experiments.

Our results, shown in Fig. 1 and Datasets EV1 and EV2, validate our experimental approach. First, the experiments interrogating RNA binding proteins (Fig. 1C) show that the "no RNA" bait experiment pulls down mainly the ACACA (Acetyl-CoA carboxylase 1) protein, which is a biotin carboxyl carrier (Dataset EV2), while Gene Ontology (GO) analysis of the RNA-5′-P putative binding proteins shows a clear enrichment in RNA binding proteins (*P* value of 1.2E-15). Second, the experiments interrogating 5′-Pme RNA 'reader' proteins (Fig. 1D), reveal proteins whose binding is either enhanced or inhibited by the modification. XRN1 and XRN2 are among the proteins whose binding is inhibited by 5′-Pme. These proteins are both 5′-P-dependent RNA exonucleases of RNAs with 5′ mono-phosphate ends (Nagarajan et al, 2013). This is in agreement with our previous results showing that the activity of Terminator, a commercial version of yeast Xrn1, is completely inhibited by the presence of 5′-Pme (Xhemalce et al, 2012). Interestingly, among the 5′-Pme binding proteins, there is a clear and reproducible enrichment of a set of proteins that are members of the multisynthetase complex, which is comprised of a bifunctional glutamyl-prolyl-tRNA synthase (EPRS), the mono-specific isoleucyl (IARS), leucyl (LARS), glutaminyl (QARS), methionyl (MARS), lysyl (KARS), arginyl (RARS) and aspartyl (DARS) tRNA synthetases, and three auxiliary proteins: EEF1E1/AIMP3/p18, AIMP2/p38 and AIMP1/p43 (Figs. 1D and 2A, B; Dataset EV1). Given that the multisynthetase complex plays crucial roles in cells, through both its canonical function of tRNA charging, and its noncanonical roles in the cellular response to DNA damage (Park et al, 2005), interferon γ (Sampath et al, 2004b), viral infection (Lee et al, 2016), and metabolic status (Arif et al, 2017), we decided to investigate further.

### The EPRS subunit of the multisynthetase complex preferentially binds to 5′-Pme RNAs

The fact that all the members of the multisynthetase complex were pulled down by the 5′-Pme RNA does not mean that all of them directly recognize the modification (Bartke et al, 2010). It is likely that one of the proteins directly recognizes the RNA modification (the direct binder), while the other proteins simply are in the same complex as the direct binder (passenger binders) (Fig. 1A). In order to identify which subunit of the multisynthetase complex directly binds to the 5′-Pme modification, we tested the binding of each of the proteins of the multisynthetase complex separately to our biotinylated RNA probes in vitro under conditions similar to the ChemRAP assay (Fig. 2C). The majority of the multisynthetase complex proteins either did not bind to any of the two RNA baits or showed weak binding to the 5′-P RNA bait. However, there were two exceptions; MARS, which bound to both 5′-P and 5′-Pme RNAs equally without discriminating against the 5′-end modification, and EPRS, which exhibited a slight preference for the 5′-Pme RNA even under our saturating pull-down conditions (Fig. 2C). When we substituted the biotinylated miRNA duplex with the corresponding pre-miRNA hairpin (Xhemalce et al, 2012), we observed a significant preference of EPRS (Fig. 2D; Appendix Fig. S2A) but not MARS (Appendix Fig. S2B) for the 5′-Pme RNA in GST-EPRS and GST-MARS pull-down assays. These results suggested that EPRS binding to RNA is enhanced by 5′-Pme, while MARS binding is not. To further validate that EPRS binds the 5′-Pme-methylated RNA directly, we performed ultraviolet (UV)

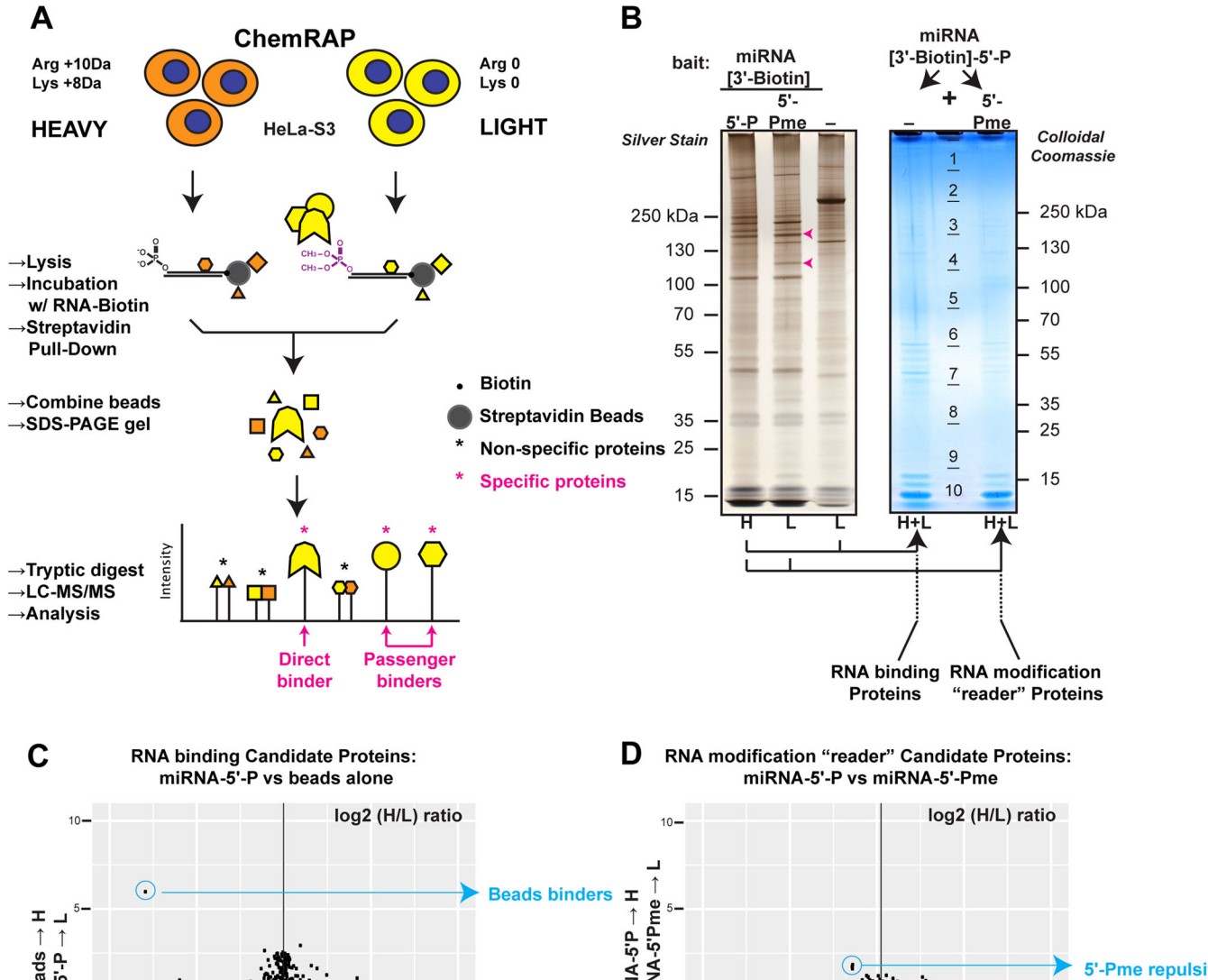

cross-linking experiments (Fig. 2E,F). After the RNA binding assay, the solution was irradiated with UV, migrated in a denaturing SDS-PAGE gel, transferred onto a PVDF membrane, and the RNA was then detected with a specific radiolabeled probe (please note that the RNA could not be directly end-radiolabeled due to the 5′-Pme modification). This assay confirmed our previous results (Fig. 2D) and verified that the bound RNA co-migrates with EPRS (Fig. 2E). Furthermore, we found that the domain of EPRS binding to 5′-Pme RNA is the non-catalytic linker region of EPRS (aa 683–1024), between the Glutamate and Proline tRNA synthetase domains

(Fig. 2F). This result is exciting because the EPRS linker is the central hub coordinating the noncanonical functions of EPRS and the multisynthetase complex (Kwon et al, 2019). In particular, the EPRS linker does not bind to tRNAs, but was previously shown to bind hairpin structures within the 3′-UTR of ceruloplasmin and VEGF-A mRNAs as part of the interferon-gamma (IFN-γ) activated inhibitor of translation (GAIT) system that dampens inflammation upon IFN-γ pathway stimulation (Arif et al, 2018; Sampath et al, 2004a). More specifically, EPRS first dissociates from the MSC and associates with NSAP1 to form the pre-GAIT

**Figure 1.  ChemRAP identifies 5′-Pme RNA modification readers.**

(A) ChemRAP experimental design for the identification of RNA modification "reader" proteins. HeLa-S3-FlpIn cells are grown in media containing either "heavy" or "light" arginine and lysine (see "Methods"). In this schematic, lysates from cells grown in "heavy" media were incubated with miRNA-5′-P [3′-Biotin], while lysates from cells grown in "light" media were incubated with miRNA-5′-Pme [3′-Biotin]. The pulldowns are pooled, resolved on a gradient PAGE gel, and subjected to in-gel trypsin digestion. The incorporation of heavy amino acids results in a mass shift of the peptides coming from the pulldowns with miRNA-5′-P [3′-Biotin]. The ratio of peak intensities in the mass spectrum reflects the relative protein abundance: Proteins that bind to the beads or the parts of RNAs other than the 5′-P end should be equally represented in both conditions and have a ratio of ~1, while the protein(s) interacting specifically with 5′-Pme should have a ratio significantly inferior to 1. (B) Left: Image of a representative silver-stained PAGE gel with 10 µL of pulldowns with either miRNA-5′-P [3′-Biotin], miRNA-5′-Pme [3′-Biotin] or "no RNA" baits. Right: Image of the corresponding Colloidal Coomassie stained PAGE gel with the indicated mixed pulldowns. NB: H heavy, L light. Red arrows point to the most prominent bands specifically observed in the miRNA-5′-Pme [3′-Biotin] pulldown. These proteins correspond to EPRS and MARS from top to bottom. (C) Plot showing the normalized log2 (H/L) ratio of a forward (*x* axis) and Reverse (*y* axis) experiment aiming at identifying miRNA binding proteins. Here, in the forward experiment, miRNA-5′-P pulldowns are with "heavy" lysates, and in the Reverse experiment with "light" lysates. Putative RNA binder proteins found in the lower right quadrant are circled in red, while beads binder proteins found in the upper left quadrant are circled in blue. (D) Plot showing the normalized log2 (H/L) ratio of a forward (*x* axis) and reverse (*y* axis) experiment aiming at identifying 5′-Pme "reader" proteins. Here, in the forward experiment, miRNA-5′-Pme pulldowns are with "heavy" lysates, and in the Reverse with "light" lysates. Putative 5′-Pme binders found in the lower right quadrant are circled in red, while 5′-Pme repulsive proteins found in the upper left quadrant are circled in blue. NB: The black lines show the median of each experiment. Full identity of proteins, with H/L ratios in forward and reverse experiments, as well as their distance to the median are found in the Datasets EV1, EV2. Source data are available online for this figure.

complex that does not bind RNA (2–4 h after IFN-γ stimulation) (Arif et al, 2009; Jia et al, 2008). Later, the formation of the complete GAIT complex, i.e., the association with GAPDH and RPL13a, as well as the rearrangement of the EPRS-NSAP1 interaction, allows binding of the EPRS linker domain to the GAIT element in 3′-UTR of specific mRNAs to inhibit their translation (16–24 h after IFN-γ stimulation) (Arif et al, 2009; Jia et al, 2008).

## BCDIN3D interacts with EPRS and regulates its association with a subset of the multisynthetase complex

In parallel to our ChemRAP experiments, we performed the analysis of BCDIN3D interacting proteins through the use of HeLa-S3-FlpIn cells with a BCDIN3D-FLAG (BCDIN3Df) integration at a single FRT locus. Interestingly, these experiments revealed that BCDIN3Df interacts in cells with MARS and EPRS and other members of the multisynthetase complex (Fig. 3A; Appendix Fig. S3). Furthermore, treatment with RNase A disrupted the interaction of BCDIN3D with both MARS and EPRS, suggesting that these interactions are mediated by RNA (Fig. 3A). Given that BCDIN3D is the 5′-Pme writer (Xhemalce et al, 2012) and that EPRS directly interacts with 5′-Pme-modified RNA (Figs. 1 and 2), we tested in vitro the interaction between EPRS and BCDIN3D in the absence or presence of RNA, with either 5′-P or 5′-Pme ends. Interestingly, we found that EPRS and BCDIN3D interact weakly in vitro in the absence of RNA, but that their binding is significantly increased specifically in the presence of 5′-Pme RNA (Fig. 3B; Appendix Fig. S2A). In contrast, BCDIN3D and MARS interact extremely weakly in vitro and their interaction remains weak in the presence of RNA regardless of its modification status (Appendix Fig. S2B, Fig. 3B and its corresponding longer exposure blot in Appendix Fig. S2C). Altogether, our results suggest that EPRS interacts with BCDIN3D and that their interaction is mediated by 5′-Pme RNA.

We next sought to assess how BCDIN3D impacts EPRS' interaction with the other subunits of the multisynthetase complex. To this end, we performed LC-MS/MS analysis of FLAG-tagged EPRS (EPRSf) in HeLa-FlpIn control and BCDIN3D knockout (BCDIN3D-KO) cells (Fig. 3C). In these experiments, BCDIN3D-KO cells showed a small (~20–25%) but reproducible defect in the interaction of EPRSf with MARS, as well as AIMP2 (Fig. 3C). This effect is also observed in immunoprecipitation (IP) of endogenous

EPRS utilizing a specific EPRS antibody in HeLa-FlpIn control and BCDIN3D-KO cells (Fig. 3D). This effect is not due to IFN-γ pathway stimulation in BCDIN3D-KO cells as EPRSf does not interact with NSAP1, RPL13A or GAPDH above background in these cells (Fig. 3E). To directly test if 5′-Pme RNA affects EPRS interaction with MARS, we performed an in vitro interaction assay, in which GST-MARS was first bound to Glutathione beads; then to mock, or RNA with 5′-P or 5′-Pme ends; and lastly to EPRS. Interestingly, we found that MARS and EPRS interact very weakly in vitro in the absence of RNA, but that their binding is most significantly increased in the presence of the 5′-Pme-modified RNA (Fig. 3F). Our findings in Fig. 3C–F suggest that the interaction between MARS and EPRS may involve a 5′-Pme RNA component in at least a sub-fraction of multisynthetase complexes in cells. This 5′-Pme RNA is not tRNA^His, as neither EPRSf or MARSf interact with tRNA^His, while they interact with both tRNA^Met and tRNA^Glu in the same Co-Immunoprecipitation conditions as in Fig. 3C–E (Fig. 3G).

## BCDIN3D regulates the interaction of specific mRNAs with EPRS

Our ChemRAP and subsequent interaction assays showed that EPRS can interact in vitro with a 5′-Pme microRNA duplex or a 5′-Pme precursor microRNA (Appendix Fig. S2) corresponding to the first discovered 5′-Pme target of BCDIN3D (Xhemalce et al, 2012). However, this does not necessarily mean that RNAs interacting with EPRS are precursor microRNAs in cells. While BCDIN3D has been shown to methylate tRNA^His (Martinez et al, 2017; Reinsborough et al, 2019) and a few specific precursor microRNAs (Reinsborough et al, 2021; Xhemalce et al, 2012), the full spectrum of BCDIN3D RNA targets is not known due to technical challenges for enrichment and detection of 5′-Pme RNA modification. In order to uncover which RNAs interact with EPRS in a BCDIN3D-dependent manner, we performed individual-nucleotide resolution UV-cross-linking and immunoprecipitation followed by next-generation sequencing (iCLIP-seq) (Huppertz et al, 2014) in HeLa-S3-FlpIn control and BCDIN3D-KO cells using the same anti-EPRS antibody as in Fig. 3D. We chose to continue performing our mechanistic investigations in the HeLa-S3-FlpIn cell line because, unlike MDA-MB-231 cell line in which BCDIN3D knockout is lethal and its knockdown leads to a defect in global

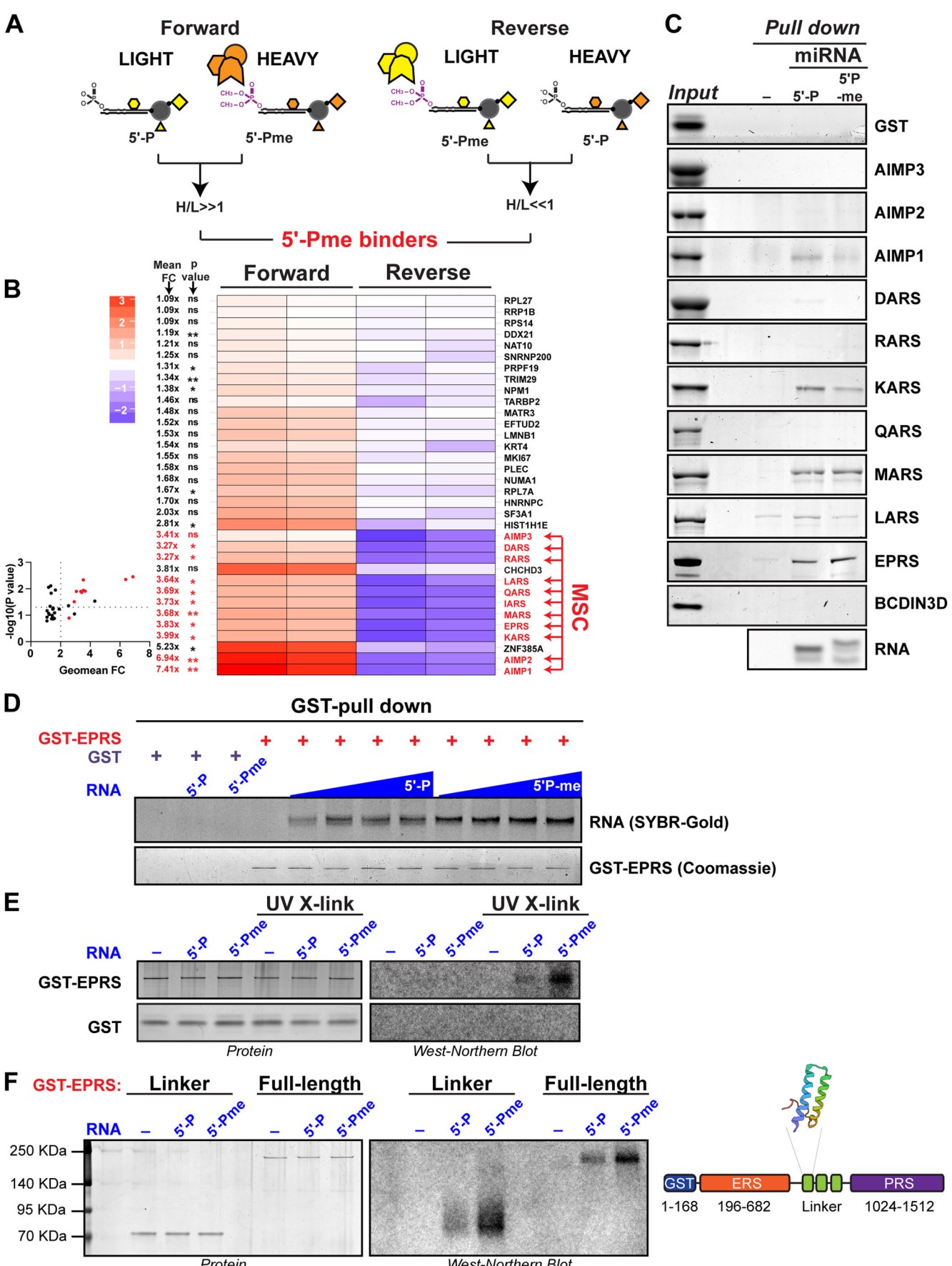

Figure 2. The EPRS subunit of the multisynthetase complex preferentially binds to 5′-Pme RNAs.

(A) Schematic of a set of forward and reverse ChemRAP experiments for the identification of 5′-Pme binder proteins. (B) Left: Heatmap cluster analysis showing the normalized log2 (H/L) ratio of all forward and reverse experiments focused on 5′-Pme binding proteins, i.e., with a $\log_2$(H/L) < 0 in the reverse experiment, and $\log_2$(H/L) > 0 in the forward experiment. On the left, shown are the mean fold change (Mean FC) of the 5′-Pme/5′-P ratio of each protein binding, as well as the associated $P$ value (multiple ratio $t$ test). The arrows highlight the multisynthetase complex subunits. Right: Volcano plot showing the geometric mean FC on the $x$ axis and the −log10($P$ value) on the $y$ axis. The red dots highlight the multisynthetase complex subunits. (C) Coomassie Stain analysis of in vitro pulldowns of the indicated recombinant proteins with either "no RNA", miR-145-5′-P [3′-Biotin], or miR-145-5′-Pme [3′-Biotin]. SYBR stain analysis of the RNA baits pulled down with Streptavidin beads is also shown as a control. (D) GST pulldown with GST or GST-EPRS of pre-miRNA-145-5′-P or pre-miRNA-145-5-Pme. The pulldown with GST is with 240 nM of RNA, while the gradients are twofold increases from 30 to 240 nM. (E) GST pulldown as in (D), followed by UV cross-linking and analysis of bound pre-miR-145 with West-Northern blot. (F) GST pulldown as in (E) using either the linker region (aa 683–1024) of EPRS or full-length GST-EPRS. Source data are available online for this figure.

translation likely due to reduced mTOR signaling (Reinsborough et al, 2021; Appendix Fig. S4), BCDIN3D knockout is viable and does not lead to a visible defect in global translation in HeLa-S3-FlpIn cells (Appendix Fig. S5). In the iCLIP procedure, the cells are UV-irradiated leading to the formation of crosslinks between proteins and their interacting RNA. The cells are lysed, and after partial RNA digestion, the protein/RNA complexes are immunoprecipitated, with high salt washes of immunoprecipitates ensuring disruption of protein–protein interactions. During library preparation, the immunoprecipitated RNA is dephosphorylated, a 3′-end adapter is ligated and the 5′-end is radioactively labeled. The protein/RNA complexes are then separated by SDS-PAGE. A representative image of the autoradiogram of the membrane resulting from this stage of the iCLIP-seq library preparation is shown on Fig. 4A. This image clearly shows that EPRS, which migrates at ~170 KDa, interacts with RNA. Moreover, BCDIN3D-KO severely decreases the levels of RNAs directly interacting with EPRS (Fig. 4A, bottom graph).

We performed our iCLIP-seq libraries from RNAs recovered from the portion of the membrane indicated by the bracket on Fig. 4A (RNase + samples only). Upon Illumina sequencing of our iCLIP-seq libraries, we analyzed our data with two different pipelines; one focused on small RNAs and one on the whole transcriptome (Fig. 4B and "Methods"). Our small RNA pipeline did not uncover any significant differences between HeLa-S3-FlpIn control and BCDIN3D-KO cells in the binding of EPRS to the cognate $tRNA^{Glu}$ aminoacylated by its ERS domain (Figs. 2F and 4B). This result suggests that BCDIN3D may not regulate EPRS's canonical function of tRNA charging, which is consistent with the lack of a visible defect in global translation in HeLa-S3-FlpIn-BCDIN3D-KO cells compared to control cells (Appendix Fig. S5). Unfortunately, our iCLIP-seq did not detect binding of EPRS to cognate $tRNA^{Pro}$ aminoacylated by its PRS subunit (Figs. 2F and 4B). Therefore, we cannot make definitive conclusions from the fact that we did not detect any specific binding of EPRS to $tRNA^{His}$ or to validated microRNAs, which are known targets of BCDIN3D. Nevertheless, our iCLIP-seq data is consistent with the absence of interaction with $tRNA^{His}$ observed in our Co-immunoprecipitation experiments (Fig. 3G). Interestingly, we detected binding of EPRS to a small number of mRNAs (Fig. 4B; Appendix Table S1), even in the absence of IFN-γ stimulation (Fig. 3E). The numbers of mRNA reads in HeLa-S3-FlpIn control are ~fourfold higher compared to the mRNA reads in BCDIN3D-KO cells (Fig. 4B), which is consistent with the results of the autoradiogram of the iCLIP-seq (Fig. 4A). Although the numbers of mRNAs bound by EPRS are limited, we observed that EPRS binds to coding exons of mRNAs rich in codons decoded by tRNAs charged by tRNA synthetases of the multisynthetase complex. The most extreme example is the one of the

EPRS footprint on the CANX gene, which encodes for APQPDV-KEEEEEKE protein sequence (Appendix Fig. S6). Given the defect in the interaction of MARS with EPRS in BCDIN3D-KO cells (Fig. 2C), we were particularly intrigued by the fact that in several mRNAs, the EPRS footprint overlaps with the start codon, which is decoded by the initiator $tRNA^{Met}$ aminoacylated by MARS (Fig. 3C; Appendix Fig. S6). One of these mRNAs with an EPRS footprint overlapping with the start codon is LRPPRC (leucine-rich pentatricopeptide repeat containing), a nuclear-encoded mitochondrial protein key for mitochondrial translation and associated with the French Canadian Leigh syndrome (LSFC) (Cui et al, 2019). As its name indicates, this protein is highly enriched with leucine amino acids (~14%) charged onto tRNAs by the LARS subunit of the multisynthetase complex. Given the association of the BCDIN3D locus with obesity and type II diabetes (Berndt et al, 2013; Reinsborough et al, 2021; Thorleifsson et al, 2009; Walley et al, 2009), we decided to focus our mechanistic efforts on this mRNA.

## BCDIN3D caps the 5′-end of the LRPPRC mRNA

As shown on Fig. 4D, we were able to validate our iCLIP-seq results with X-RIP which uses formaldehyde cross-linking instead of UV (please note that BCDIN3D-KO does not affect the levels of LRPPRC mRNA, Dataset EV3 and Appendix Table S2). While inspecting the 5′-UTR of LRPPRC mRNA, we were intrigued by its predicted hairpin secondary structure and the presence of two stretches of five nucleotides identical in sequence and similar in positioning to $tRNA^{His}$ (Appendix Fig. S7). Therefore, we hypothesized that LRPPRC mRNA may be a target of BCDIN3D-mediated methylation. Unfortunately, there is no available antibody to enrich 5′-Pme-modified RNAs, and mass spectrometry and differential gel migration methods used to detect 5′-Pme in $tRNA^{His}$ are not amenable to mRNAs due to differences in length and abundance (Devanathan et al, 2021). To overcome these technical limitations, we used the property of 5′-Pme-modified RNAs of being resistant to Terminator treatment (Xhemalce et al, 2012). As mentioned earlier, Terminator corresponds to yeast Xrn1, which exclusively degrades RNAs with 5′-P ends, leaving intact RNAs with $m^7G$, 5′-PPP, 5′-Pme or 5′-OH caps/ends (Fig. 5A). To render RNAs with $m^7G$ caps or 5′-PPP ends sensitive to Terminator, we treated total RNA with TAP (Tobacco Acid Pyrophosphatase), which converts $m^7G$ and 5′-PPP RNAs and their derivatives into 5′-P (Fig. 5A). As shown on the Bioanalyzer image on Fig. 5B, our treatment with Terminator of total RNA resulted in efficient degradation of 28 S, 18 S, and 5.8 S rRNAs that all have 5′-P ends, but not 5 S rRNA that has a 5′-PPP end, as expected and observed previously (Xhemalce et al, 2012). When the total RNA is pre-treated with TAP, 5 S RNA is also degraded by Terminator (Fig. 5B). The same is true for the 7SK and U4 RNAs with 5′-PPPme and TMG caps, respectively (SYBR-Gold-stained gel on

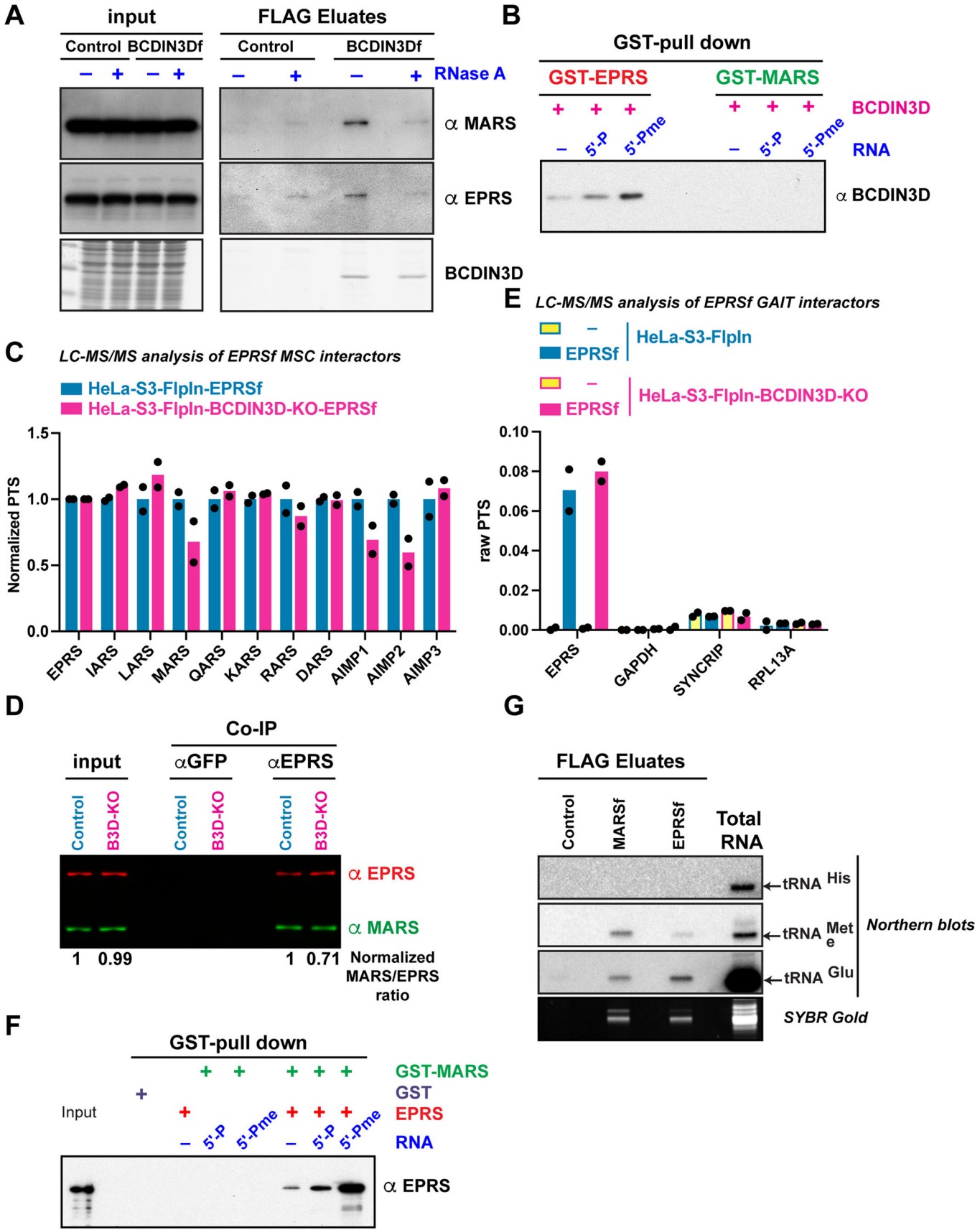

◀ **Figure 3.  BCDIN3D interacts with EPRS and regulates its association with a subset of the multisynthetase complex.**

(**A**) HeLa-S3-FlpIn Control and BCDIN3Df (BCDIN3D-FLAG) lysates were treated with mock or 30 µg RNase A prior to FLAG co-immunoprecipitation and elution with a FLAG peptide. Inputs and FLAG eluates were analyzed by western blots with the indicated antibodies. Equal co-immuno-precipitation of BCDIN3D was verified by Coomassie staining. (**B**) Direct comparison of BCDIN3D binding to GST-EPRS and GST-MARS (See complete analysis in Appendix Fig. S2). (**C**) FLAG eluates from HeLa-S3-FlpIn ± EPRSf (EPRS-FLAG) or HeLa-S3-FlpIn-BCDIN3D-KO ± EPRSf were analyzed by LC-MS/MS. Plotted is the mean from $n = 2$ independent biological replicates of the Percentage of Total Spectra (PTS) for each protein normalized to EPRS PTS and HeLa-S3-FlpIn-EPRSf. (**D**) Quantitative LI-COR western blot analysis with antibodies against EPRS (red) and MARS (green) of input and anti-GFP or anti-EPRS immune-precipitates of HeLa-S3-FlpIn Control and BCDIN3D-KO cells. Below the western blot is shown the ratio of MARS/EPRS normalized to control. (**E**) Enrichment of GAIT subunits (EPRS, GAPDH, NSAP1 and RPL13A) in HeLa-S3-FlpIn and HeLa-S3-FlpIn-BCDIN3D-KO: -(Control) and -EPRSf FLAG eluates. Plotted is the mean from $n = 2$ independent biological repeats of the PTS for each protein. (**F**) GST pulldown with GST-MARS assessing binding of untagged recombinant EPRS in the absence or presence of RNA-5′-P or 5′-Pme. (**G**) Northern blot analysis of MARSf and EPRSf interacting RNAs. The bottom panel shows the SYBR-Gold-stained gel used for the northern blots on the top. Source data are available online for this figure.

Fig. 5B), and the B2M control mRNA (Fig. 5C). At the difference of B2M, around 25% of the LRPPRC mRNA remains resistant to Terminator in control cells (Fig. 5C), indicating that the LRPPRC mRNA contains a noncanonical 5′ cap. Given that the ratio of Terminator-resistant LRPPRC is reduced in BCDIN3D-KO cells compared to control (Fig. 5C), this cap is likely 5′-Pme.

To validate this finding, we performed in vitro RNA methyl-transferase assay with LRPPRC mRNA and recombinant BCDIN3D. We found that, in vitro, BCDIN3D cannot methylate an RNA with a 5′-P end that corresponds strictly to the 5′-UTR of LRPPRC, however its activity towards the 5′-P is substantially increased when the RNA is simply extended with the downstream sequence of the LRPPRC mRNA which is predicted to form a double-stranded structure where the two first nucleotides are unpaired but the next 8 are paired with complementary sequences downstream of the start codon (Fig. 5D) as is the case in tRNA[His] (Liu et al, 2020). Taken together, data obtained from the TAP ±Terminator treatment assay (Fig. 5A–C) and in vitro RNA methyltransferase assay (Fig. 5D) suggest that a portion of the LRPPRC mRNA is capped with a 5′-Pme by BCDIN3D.

## BCDIN3D reduces translation initiation of LRPPRC mRNA

Given that BCDIN3D does not affect LRPPRC mRNA levels (Appendix Table S2), we sought to determine whether BCDIN3D affects LRPPRC protein instead. We first analyzed whole-cell extracts from HeLa-S3-FlpIn control and BCDIN3D-KO cells with quantitative LI-COR western blot using a specific LRPPRC antibody (Fig. 6A). This analysis showed a ~1.5-fold increase of LRPPRC protein in BCDIN3D-KO compared to control (Fig. 6B). In order to check whether the increase in protein levels was due to increased translation, we analyzed the distribution of the LRPPRC mRNA in polysome fractions (Fig. 6C; Appendix Fig. S5). The LRPPRC mRNA showed a pattern very different from the B2M control mRNA in both control and BCDIN3D-KO cells (Fig. 6C; Appendix Fig. S5). While the vast majority of the B2M mRNA was located in the polysome fractions, the LRPPRC mRNA accumulated in monosome/high molecular weight RNP fractions, consistent with highly regulated translation initiation and less efficient translation. In BCDIN3D-KO cells, the LRPPRC mRNA shifted to lighter molecular weight RNP fractions, but higher molecular weight polysome fractions, consistent with higher translation rates of the LRPPRC mRNA in BCDIN3D-KO versus control cells. In order to check whether this effect is due to higher rates of initiation, we performed Ribo-seq in cells pre-treated with harringtonine, an initiation-specific translation inhibitor, that halts ribosomes at

initiation codons (Ingolia et al, 2012). This experiment clearly showed an increased presence of ribosome-protected fragments (RPFs) in BCDIN3D-KO versus control cells (Fig. 6D).

This result prompted us to probe whether BCDIN3D affects LRPPRC expression via its 5′UTR. We noticed that the LRPPRC mRNA has a short 5′UTR, followed by two in frame AUG codons (AUG#1 and AUG#2), with the first AUG (AUG#1) being immediately followed by the sequence coding for a 58 aa-long transit peptide responsible for LRPPRC transport into the mitochondrial matrix (Fig. 6E; Appendix Fig. S6). Thus, we engineered an LRPPRC-5′UTR-GFP reporter (Fig. 6E), containing the LRPPRC mRNA sequence down to AUG#2 fused to GFP. This sequence was cloned into a pcDNA5-FRT vector that can be inserted at the single FRT locus to allow for equal transcription of the reporter in both HeLa-S3-FlpIn control and BCDIN3D-KO cells. Upon engineering the LRPPRC-5′UTR-GFP reporter in both HeLa-S3-FlpIn control and BCDIN3D-KO cells, we performed flow cytometry to precisely quantify the levels of LRPPRC-5′UTR-GFP reporter proteins in these cell lines. Our analysis showed that BCDIN3D-KO cells have significantly higher GFP intensity than control cells (Fig. 6F, mean of $7 \times 10^6$ AU in BCDIN3D-KO versus $3.7 \times 10^6$ AU in control GFP$^+$ cells). These experiments demonstrate that the 5′UTR of LRPPRC mRNA is sufficient to confer more translation to the GFP reporter in BCDIN3D-KO cells.

## BCDIN3D may regulate local translation of LRPPRC mRNA

To also investigate the effect of BCDIN3D knockout on LRPPRC protein localization, we performed both cellular fractionation and mitochondrial enrichment of HeLa-S3-FlpIn control and BCDIN3D-KO cells and analyzed endogenous LRPPRC as well as EPRS and MARS proteins with quantitative western blotting.

We first performed cellular fractionation into five fractions: cytoplasm, membrane, nuclear, chromatin-bound, and cytoskeleton-bound, as previously described (Shelton et al, 2018). We observed that EPRS and MARS display very similar patterns to each other in control cells, i.e., fractionating at ~50% in the cytoplasm, ~30% in membranes, and ~20% in nucleosol (Appendix Fig. S8). In BCDIN3D-KO cells, EPRS but not MARS displayed a significant increase in its membrane-bound fraction (Appendix Fig. S8). In these same experiments, we observed that LRPPRC protein showed a mostly membrane-bound fractionation, which is expected as the membrane fraction contains mitochondria in addition to dissolved plasma and ER/Golgi membranes. Compared to control, BCDIN3D-KO cells also displayed a small but significant increase

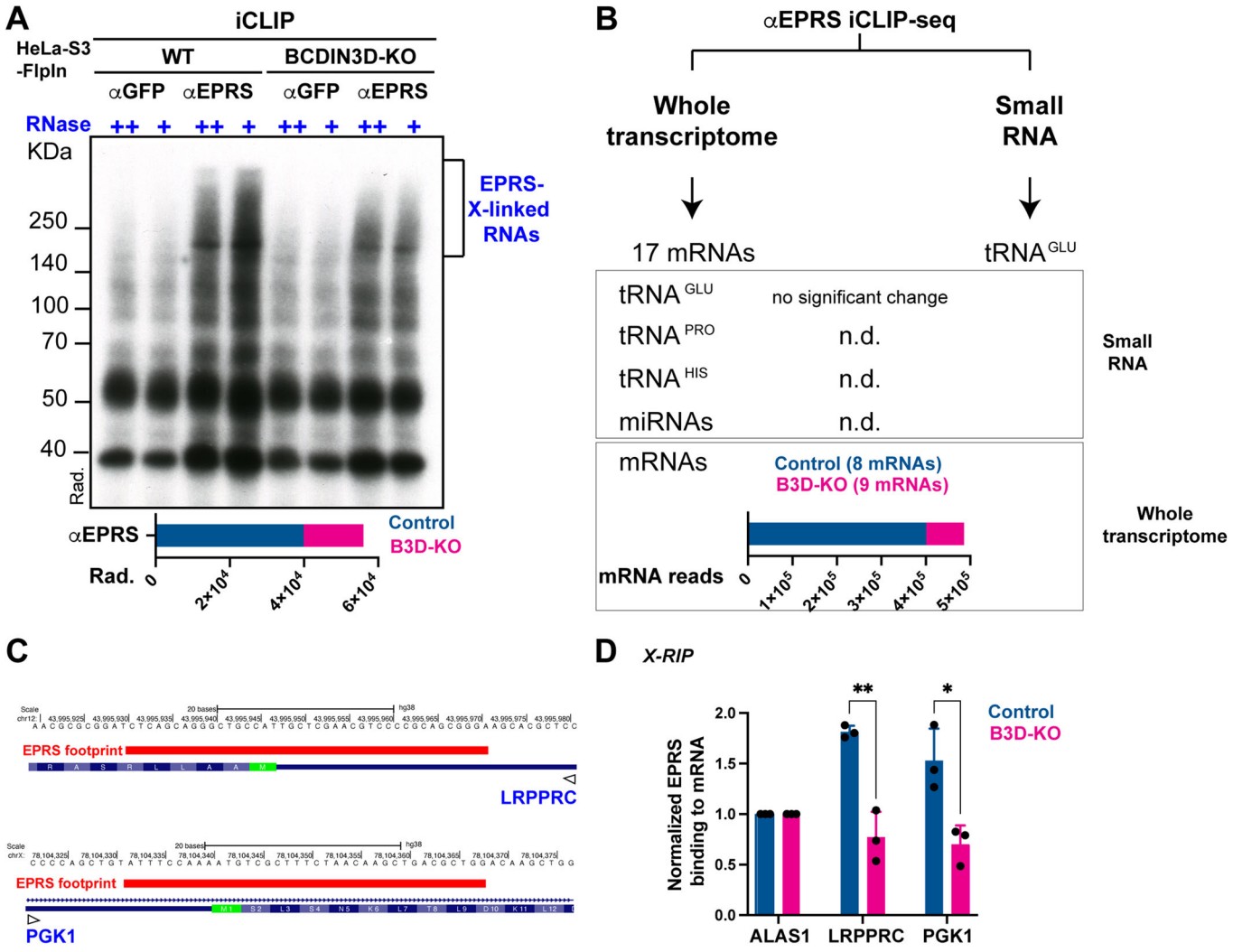

**Figure 4. BCDIN3D regulates interaction of specific mRNAs with EPRS.**

(A) Autoradiogram of the membrane stage of the iCLIP-seq library preparation showing migration of protein-RNA crosslinks pulled down by the control (anti-GFP) and anti-EPRS antibodies in HeLa-S3-FlpIn control and BCDIN3D-KO cells. The bracket shows the EPRS-cross-linked RNAs and corresponds to part of the membrane that was recovered to subsequently perform the iCLIP-seq. The bottom graph shows the quantification of the radioactivity incorporated in the anti-EPRS (RNase +) samples used for the iCLIP-seq. (B) Summary of iCLIP-seq results in HeLa-S3-FlpIn control and BCDIN3D-KO cells as in (A) analyzed by two different pipelines, whole transcriptome analysis and small RNA analysis. The results specific to anti-EPRS compared to anti-GFP for RNAs of interest are shown (n.d. stands for "not detected"). For mRNAs, raw mRNA read numbers pooled from two iCLIP-seq repeats are shown. (C) iCLIP-seq EPRS footprints on the LRPPRC and PGK1 mRNAs shown on UCSC genome browser (hg38). For each example, shown are: the scale, the position on the chromosome, the DNA sequence of the Watson strand (note that the coding sequence of LRPPRC gene is on the Crick strand), the EPRS footprint, the representation of the gene with thin lines representing introns, thick lines representing coding exons [with the encoded Methionines (M) in green and other amino acids in blue], and intermediate thickness lines representing UTRs. (D) Validation of iCLIP-Seq results by X-RIP-RTqPCR with control (anti-GFP) and anti-EPRS antibody in HeLa-S3-FlpIn control and BCDIN3D-KO cells. Shown are the levels of LRPPRC and PGK1 mRNAs normalized to input, GFP and ALAS1 control gene (mean ± SD, n = 3 independent biological replicates, *P value < 0.05, **P value < 0.01 in multiple unpaired t test). Source data are available online for this figure.

in the membrane-bound fraction of LRPPRC protein (Appendix Fig. S9).

To more precisely probe the association of LRPPRC with mitochondria, we enriched mitochondria from HeLa-S3-FlpIn control and BCDIN3D-KO cells using a differential centrifugation method (see "Methods"). We observed that localization of LRPPRC to the mitochondria-enriched fraction is significantly decreased in BCDIN3D-KO cells (Fig. 6G,H). Our data in Fig. 6A–G and Appendix Fig. S9 suggest that in BCDIN3D-KO cells, LRPPRC protein is translated more overall, but displays concomitant

decreased localization to mitochondria and increased association with another type of membrane or membraneous organelle(s). The decreased localization to mitochondria is not due to a generalized mitochondrial transport defect. Indeed, when we constructed a HeLa-FlpIn cell line carrying the LRPPRC-5′UTR-GFP reporter, we observed perfect colocalization of the GFP signal with the MitoTracker-Red signal in both siNC control and siBCDIN3D cells (Fig. 6I), showing that (i) the LRPPRC-5′UTR-GFP reporter is translated using the first AUG and gets transported to mitochondria, and (ii) that these two processes are not affected by

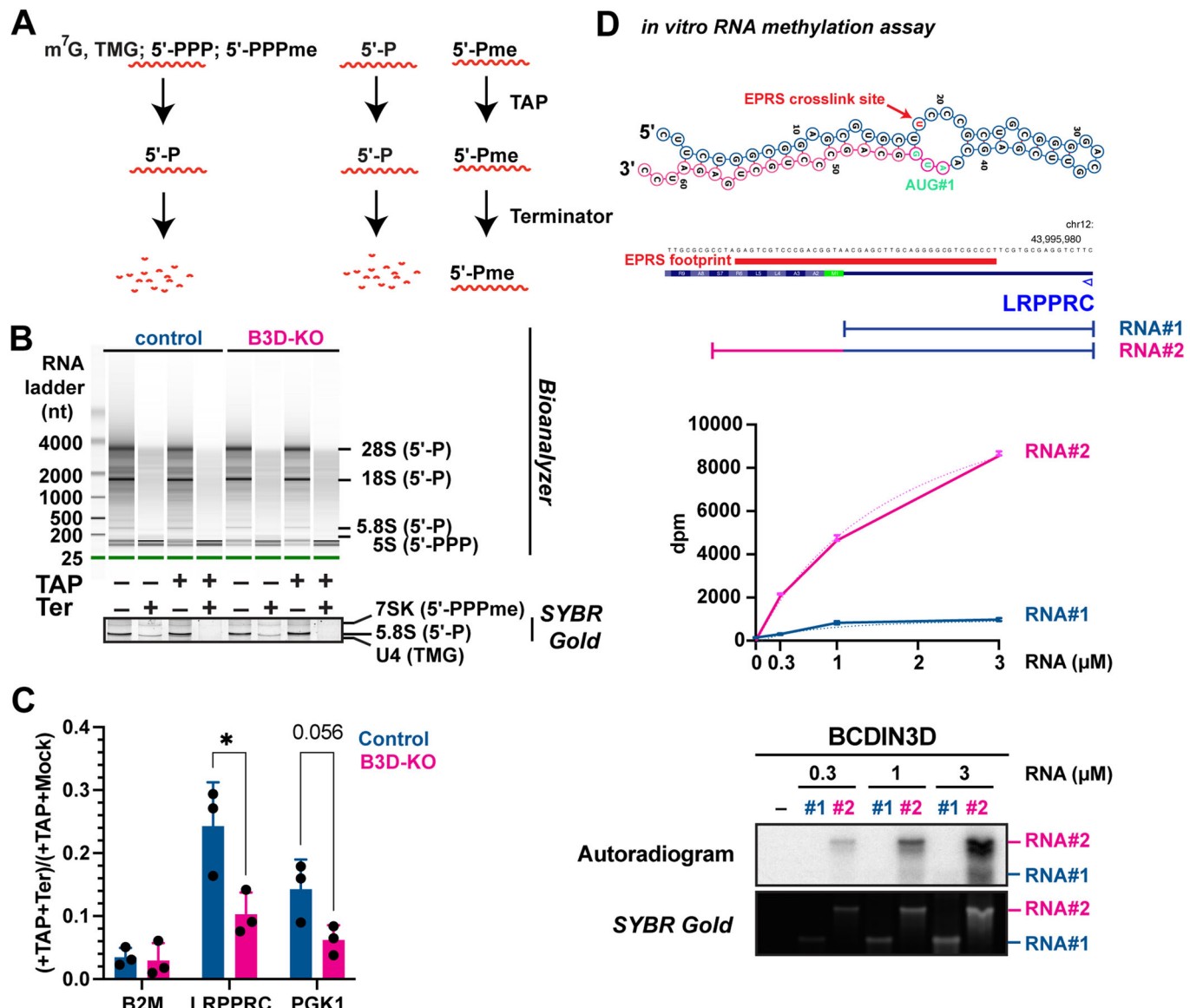

**Figure 5. BCDIN3D methylates specific mRNAs ends.**

(A) Schematic of the activity of TAP (tobacco acid pyrophosphatase) and Terminator on various 5′ ends. TMG represents TriMethylGuanosine caps. (B) Bioanalyzer traces of RNAs treated with mock or TAP and/or Terminator. Shown is also a SYBR-Gold-stained PAGE gel focused on the 7SK, 5.8 S and U4 snRNAs shown by arrows. (C) Analysis of the Terminator-resistant fraction of LRPPRC mRNA in total RNA from HeLa-S3-FlpIn Control and BCDIN3D-KO cells pre-treated with TAP by RTqPCR. Shown is the (TAP+Terminator)/(TAP+Mock) ratio of the levels of B2M, LRPPRC, and PGK1 mRNAs (mean ± SD, $n = 3$ independent biological replicates, *P value < 0.05 in a multiple unpaired $t$ test). See "RNA analysis" under "Methods", for more details. (D) In vitro RNA methyltransferase assay with BCDIN3D using radioactive [³H]-SAM as methyl group donor, and RNA#1 and RNA#2. RNA#1 corresponds to the 5′ UTR of LRPPRC (in teal color). RNA#2 corresponds to the 5′ UTR of LRPPRC extended to the open reading frame (extension sequence shown in magenta). Both RNA#1 and RNA#2 have 5′-P ends. Top: Predicted two-dimensional structure of RNA#2, with the sequence in common with RNA#1 shown in teal; the sequence unique to RNA#2 shown in magenta; EPRS-cross-linked site shown with red text; and the start codon shown with green text. Middle: Scintillation counts in disintegrations per minute (dpm) of C[³H]₃ incorporated into the RNA from the RNA methyltransferase assay in the bottom panel (mean ± SD, $n = 3$ technical replicates). Bottom: The bottom panels show the autoradiography and the SYBR-Gold-stained gel that was used for the autoradiography of the RNA methyltransferase assay. Source data are available online for this figure.

BCDIN3D. Thus, the decreased localization of endogenous LRPPRC protein in BCDIN3D-KO cells may be due to intrinsic properties of the LRPPRC protein, which may require translation of its mRNA near mitochondria for proper mitochondrial transport. Consistent with this, in the same mitochondria enrichment experiments, EPRS and MARS were both associated with the mitochondria-enriched fraction at similar levels in control cells

(Fig. 6G,H), suggesting that they form a complex in proximity to mitochondria. BCDIN3D-KO did not affect EPRS fractionation with the mitochondria-enriched fraction, but it significantly decreased MARS (Fig. 6G,H). Together with the Co-IP data in Fig. 3C,D, our mitochondria enrichment data in Fig. 6F,G suggest that EPRS and MARS interaction is partially decreased at the proximity of mitochondria in BCDIN3D-KO cells.

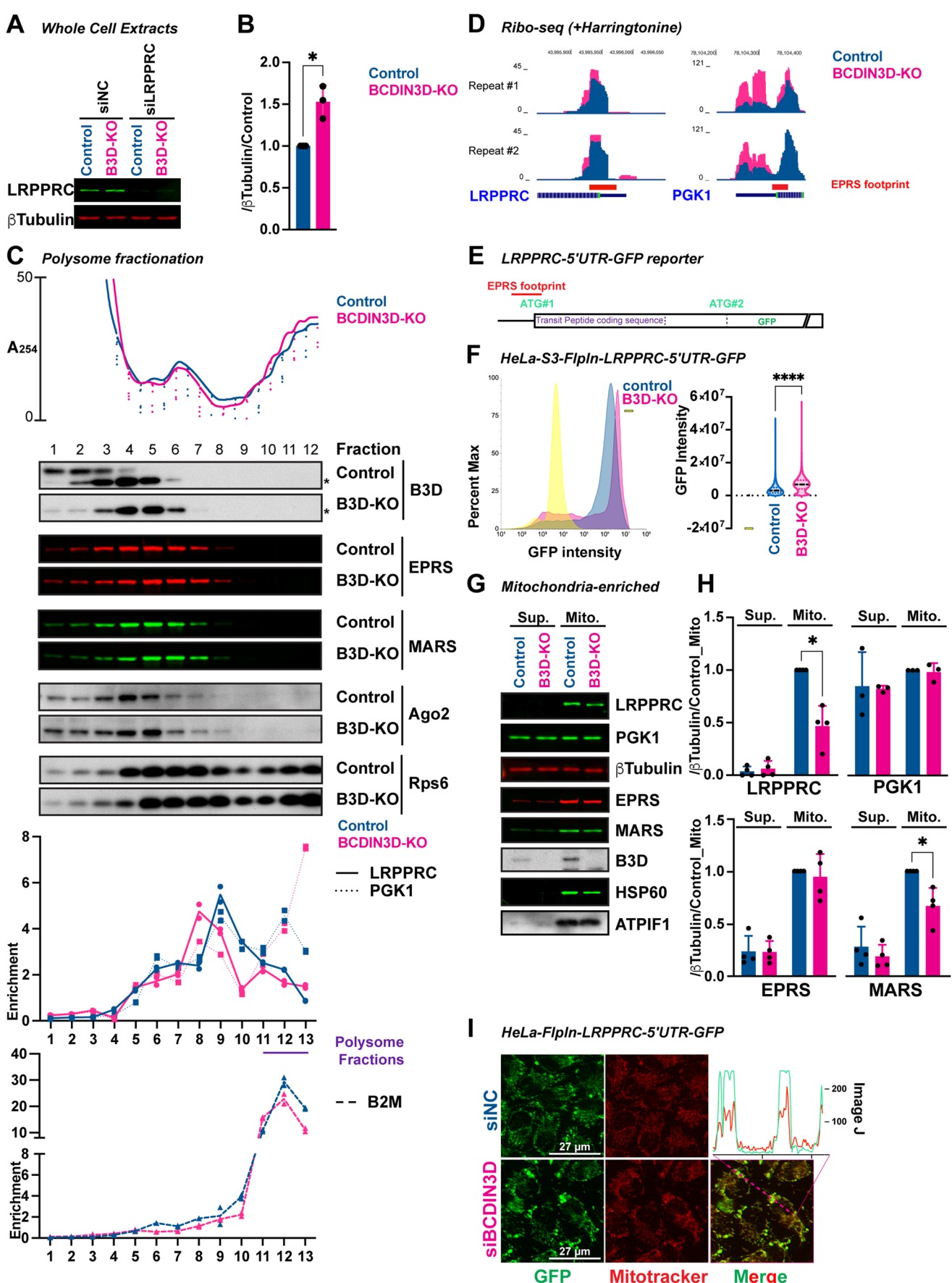

**Figure 6. BCDIN3D regulates LRPPRC translation.**

(A) Representative quantitative LI-COR western blots with antibodies against LRPPRC (green) and β-Tubulin (red) of whole-cell extracts collected after 48 h of reverse transfection of siNC and siLRPPRC siRNAs in HeLa-S3-FlpIn control and BCDIN3D-KO cells. (B) Ratio of LRPPRC over β-Tubulin normalized to control in quantitative LI-COR western blots of HeLa-S3-FlpIn control and BCDIN3D-KO whole-cell extracts (mean ± SD, $n = 3$ independent biological replicates, *$P$ value = 0.03 for LRPPRC in a paired ratio $t$ test). (C) Polysome lysates from HeLa-S3-FlpIn control and BCDIN3D-KO cells were fractionated on a 7–50% sucrose gradient and shown are from top to bottom: the real-time recording of $OD_{254}$; western blots with the indicated antibodies of 20 μL of each fraction (asterisk indicates a non-specific band detected by the BCDIN3D antibody); RTqPCR analysis of LRPPRC, PGK1 and B2M mRNA from each fraction of the same polysome fractionation (shown is mean from two technical replicates). Normalization was done over the average Ct of each mRNA, which did not show significant differences in Control and BCDIN3D-KO cells. See also Appendix Fig. S5 for more details. (D) Harringtonine-treated Ribo-seq data for the LRPPRC and PGK1 mRNAs translation initiation sites are shown on the UCSC genome browser (hg38). (E) Schematic of the LRPPRC-5′UTR-GFP reporter. (F) Flow cytometry analysis of GFP intensity in HeLa-S3-FlpIn control and BCDIN3D-KO cells with a single copy of the LRPPRC-5′UTR-GFP reporter at the FRT locus. Shown are also HeLa-S3-FlpIn cells without reporter (− in yellow) as a negative control. Left: Histogram distribution of GFP intensity (Arbitrary Units) from ~30,000 cells per sample. Right: violin plot of GFP intensity (Arbitrary Units) from ~30,000 single cells per sample. (****$P$ value < 0.0001 in one-way ordinary ANOVA with Tukey's multiple comparisons test, only the result of the control/BCDIN3D-KO pair is shown). (G) Representative quantitative LI-COR western blots of LRPPRC, PGK1, β-Tubulin, EPRS and MARS distribution upon mitochondrial enrichment in HeLa-S3-FlpIn control and BCDIN3D-KO cells. Shown are also BCDIN3D and two mitochondrial markers (ATPIF1 and HSP60). Sup. stands for supernatant, and Mito. stands for mitochondria-enriched fraction. (H) Ratio of LRPPRC, PGK1, EPRS, and MARS over β-Tubulin normalized to the control mitochondria-enriched fraction of quantitative LI-COR western blots (mean ± SD, $n = 3$–4 independent biological repeats, *$P$ value = 0.047 for LRPPRC and *$P$ value = 0.057 for MARS in a multiple paired ratio $t$ test). Sup. stands for supernatant, and Mito. stands for mitochondria-enriched fraction. (I) Left: Representative images of HeLa-FlpIn siNC and siBCDIN3D cells having a single copy of the LRPPRC-5′UTR-GFP reporter at the FRT locus and stained for 15 min with Mitotracker-Red. Right: Raw ImageJ profile analysis of the line shown on the siBCDIN3D merged image for each channel. Source data are available online for this figure.

The effect of BCDIN3D knockout with respect to increased translation is not limited to the LRPPRC mRNA, as we were able to obtain similar results with the PGK1 mRNA, which is also bound by EPRS over the start codon (Figs. 3C and 4D), displays 10 to 20% noncanonical capping by BCDIN3D (Fig. 4G) and features of higher translation (Fig. 6C). At the difference of LRPPRC, PGK1 protein levels in the cells are not upregulated (Fig. 6G,H), probably due to post-translational compensatory mechanisms. Interestingly, Ribo-seq analysis in the presence of harringtonine showed a highly increased level of RPF upstream of the PGK1 start codon in BCDIN3D-KO cells (Fig. 6D), suggesting that BCDIN3D and EPRS may protect translation initiation at upstream noncanonical sites.

## Discussion

Using ChemRAP, we identified the multisynthetase complex as a reader for the 5′-Pme RNA modification (Figs. 1 and 2). We further showed that EPRS is the subunit of the multisynthetase complex that directly binds to the 5′-Pme modification through its linker non-catalytic domain (Fig. 2C–F), and that the interaction between EPRS and MARS is specifically increased in the presence of a 5′-Pme RNA in vitro (Fig. 3F). In cells, Co-IP of FLAG-tagged or endogenous EPRS analyzed by LC-MS/MS or quantitative western blot revealed that a subset of EPRS does not interact with MARS (Fig. 3C,D). EPRS iCLIP-seq further showed that EPRS interacts with a small set of specific mRNAs, including LRPPRC, to which EPRS binds on the immediate proximity of the start codon (Appendix Fig. S6; Fig. 4A–C). Binding of EPRS to this mRNA site is abolished in BCDIN3D-KO cells (Fig. 4D). Our data are consistent with BCDIN3D directly methylating the 5′-P of the LRPPRC mRNA (Fig. 5A–D), suggesting that 5′-Pme may mediate binding of EPRS to LRPPRC-5′UTR. Furthermore, the LRPPRC-5′UTR is sufficient to cause overexpression of a GFP reporter in BCDIN3D-KO or siBCDIN3D cells (Fig. 6E,F), suggesting that EPRS binding to LRPPRC may be inhibitory. Interestingly, cellular fractionation and mitochondria enrichment showed that the EPRS-MARS interaction defect observed in BCDIN3D-KO cells is likely

occurring in proximity to mitochondria and potentially other membranes (Appendix Fig. S8; Fig. 6G,H). Based on these data, EPRS and MARS complex formation on the LRPPRC mRNA may act to inhibit its translation until it is in proximity to mitochondria. We further speculate that LRPPRC translation away from mitochondria may result in defective LRPPRC protein mitochondrial transport and/or function (Fig. S9 and Fig. 6G,H), likely due to its intrinsic structural features and/or regulation. A similar effect has recently been observed for the NET1 mRNA by the Mili lab (Gasparski et al, 2023).

Our studies present several limitations that are worth considering. As mentioned above, our results do not exclude that other RNAs methylated by BCDIN3D mediate interactions of EPRS with messenger RNAs, especially within the mRNA coding sequence (CDS), which cannot be methylated by BCDIN3D. Although we did not find evidence of such RNAs in our EPRS iCLIP-seq data, technical issues likely limited the sensitivity of our iCLIP-seq, resulting in the absence of observed interaction between EPRS and its cognate tRNA^Pro aminoacylated by its PRS subunit even in control cells. To our knowledge, this is the first unbiased analysis of EPRS RNA interactors by iCLIP-seq, and further antibody and technical improvements will likely lead to the discovery of more (m)RNAs regulated by BCDIN3D and EPRS. Future work that is reliant on the development of tools for direct detection of 5′-Pme will also allow the study of the dynamics of this modification in a spatiotemporal-resolved manner.

In addition, to avoid potential complexities in data analyses, here, we performed most of our cellular assays in suspension HeLa-S3-FlpIn cells in which BCDIN3D-KO is viable and does not lead to global translation defects. However, we cannot exclude that in other cellular contexts, BCDIN3D-dependent methylation and its RNA targets do not regulate MSC function in a more extensive manner than reported here. We have observed this to be the case in MDA-MB-231 cells, in which BCDIN3D is essential, while its knockdown leads to global translation suppression (Reinsborough et al, 2021) as well as EPRS and MARS fractionation defects in polysome fractionation experiments (Appendix Fig. S4). In addition, while we showed by both iCLIP-seq and X-RIP that LRPPRC mRNA does not bind to EPRS in BCDIN3D-KO cells, we

did not formally show whether the lack of binding of LRPPRC mRNA in BCDIN3D-KO cells is due to EPRS requiring 5′-Pme on the LRPPRC mRNA for efficient binding. Future work, based on co-crystal structure of EPRS linker domain with the 5′-Pme LRPPRC mRNA or in silico simulations, will need to identify point mutation(s) that specifically disrupt(s) the binding of EPRS linker domain to 5′-Pme. After introducing the identified EPRS mutation in the EPRS genomic locus, future work will also need to determine if the identified mutation(s) recapitulate the BCDIN3D-KO phenotype. Finally, future experiments will also be needed to determine whether BCDIN3D methylation and its RNA targets regulate other noncanonical functions of EPRS, such as the cellular responses to DNA damage (Park et al, 2005), IFN-γ (Sampath et al, 2004b), viral infection (Lee et al, 2016) and metabolic status (Arif et al, 2017). Our results linking BCDIN3D, EPRS and LRPPRC are particularly intriguing given the association of BCDIN3D locus with obesity and type II diabetes in humans (Berndt et al, 2013; Reinsborough et al, 2021; Thorleifsson et al, 2009; Walley et al, 2009), and the results from the Fox lab (Arif et al, 2017) showing that EPRS linker S999 phospho-deficient mutants have reduced adipose tissue mass, and increased lifespan.

# Methods

## Modified RNAs

miR-145*-5′-P passenger strand and 3′ biotinylated miR-145-5′-P, miR-145-5′-Pme, unbiotinylated pre-miR-145-5′-P, pre-miR-145-5′-Pme were custom synthesized by IBA GmbH. LRPPRC RNA#1 and RNA#2 were custom synthesized by Sigma and Dharmacon.

## Cell lines

HeLa-S3-FlpIn Parental and BCDIN3Df were previously described (Xhemalce et al, 2012). Other integrations in the FRT site were engineered as previously described (Xhemalce et al, 2012). All non-commercially obtained plasmids used to make the cell lines and their sequences will be made available on Addgene (https://www.addgene.org/Blerta_Xhemalce/). BCDIN3D was knocked out in HeLa and HeLa-S3-FlpIn cells with the BCDIN3D Human Gene Knockout Kit (CRISPR) from Origene (#KN208818) as recommended. The puromycin concentration used for the candidate clone selection step was at 1 μg/μL.

Non-labeled HeLa-S3-FlpIn cells were grown in spinner flasks at 75 rpm in RPMI + 10%FBS + PSQ supplemented with 200 μg/mL of Zeocin (parental) or 400 μg/mL hygromycin (BCDIN3Df, EPRSf, 5′UTR-LRPPRC-GFP reporter).

## Heavy and light cell labeling

HeLa-S3-FlpIn cell stocks were made from cells grown in "heavy" or "light" medium for 12 generations. Cells were grown in RPMI medium without lysine and arginine (PI89984), supplemented with dialyzed FBS (#26000044), either "heavy" or "light" L-arginine and L-lysine, 100 U/mL penicillin, 100 μg/mL streptomycin and 2 mM L-glutamine ["heavy": L-arginine-HCl, $^{13}C6$, $^{15}N4$ #PI88434; L-lysine-2HCl, $^{13}C6$, $^{15}N2$ #PI88432; "light": L-arginine-HCl #PI88427; L-lysine-2HCl #PI88429].

## Preparation of cytoplasmic lysates for ChemRAP

For each pulldown, $2 \times 10^7$ HeLa-S3-FlpIn cells were centrifuged for 5 min at 200 g at 4 °C, washed twice with 25 mL of ice-cold PBS and resuspended in 500 μL of CPE buffer (10 mM HEPES, pH 7.9, 10 mM KCl, 1.5 mM $MgCl_2$, 0.34 M sucrose, 10% glycerol, 1 mM DTT, 0.1% Triton X-100, supplemented with EDTA-free Complete Protease Inhibitor cocktail from Roche). Cells were incubated for 8 min on ice and centrifuged for 5 min at 1300× g at 4 °C. The supernatant, containing the cytoplasmic extracts, was collected, the concentration of KCl was adjusted to 120 mM and 5 μL of RNaseOUT were added.

## Preparation of RNAs coupled to streptavidin beads

### RNA annealing
For each pulldown, 1 μL of miR-145-[3′Biotin]-5′-P or -5′-Pme at 100 μM were mixed with 1 μL of miR-145*-5′-P at 100 μM in a total volume of 40 μL of water. The mix was heated for 2 min at 70 °C and the temperature was decreased down to 4 °C at a rate of 1 °C per min.

### RNA coupling to streptavidin beads
For each pulldown, 20 μL of magnetic streptavidin beads (Thermo-Fisher Scientific #65602) were washed three times with 20 μL of 1× binding & washing buffer (5 mM Tris-HCl pH 7.5, 0.5 mM EDTA, 1 M NaCl) and then resuspended in 40 μL of 2× buffer. The 40 μL of annealed biotinylated RNA was incubated with the beads for 15 min with gentle rotation at room temperature. The RNA-coated beads were washed three times with 100 μL of 1× binding & washing buffer, three times with 100 μL of CPE buffer (with KCl adjusted to 120 mM) prior to each pulldown, and resuspended in 100 μL of CPE buffer (with KCl adjusted to 120 mM).

## RNA pull-down and binding assays

For the ChemRAP experiment, 500 μL of cytoplasmic extracts were incubated with the 100 μL of beads for 2 h at 4 °C with rotation. The beads were washed four times with 500 μL of CPE buffer (with KCl adjusted to 240 mM) and resuspended with 50 μL of 1× Laemmli buffer.

For the validation of the ChemRAP experiments in vitro, the cytoplasmic extracts were substituted with 500 ng of purified recombinant proteins and the buffers were changed to 50 mM Tris-HCl, pH 8, 150 mM NaCl, and 0.5% NP-40.

For each in vitro RNA binding assay, 10 pmol of GST, GST-EPRS, GST-MARS proteins or mock were first bound to 25 μL of Glutathione-coupled beads. The beads were then incubated with mock or 100 pmol of synthetic RNA-5′-P or -5′-Pme in 1 mL of binding buffer 1 (50 mM Tris-HCl, pH 8, 50 mM NaCl and 0.5% NP-40) for 2 h at 4 °C. The beads were washed three times with 1 mL of binding buffer for 5 min at 4 °C. The beads were then incubated with mock or 100 pmol of EPRS-C-MYC/DDK in 1 mL of binding buffer 1 or untagged BCDIN3D in 1 mL of binding buffer 2 (50 mM Tris-HCl, pH 8, 50 mM NaCl, 0.5% NP-40 and 5 mM DTT) for 2 h at 4 °C. The beads were washed three times with 1 mL of binding buffer 1 or 2 for 5 min at 4 °C. After the last wash, the beads were split into two tubes: half was used to check protein binding as above, and half was used to extract RNA with the Qiagen RNA Cleanup purification kit as in

(Xhemalce et al, 2012) and eluted with 30 µL of water. In total, 15 µL was used for analysis on Urea-PAGE gel. The gels were stained with Silver Stain prior to scanning.

For the in vitro RNA binding assay in Fig. 2D, 10 pmol of GST and GST-EPRS proteins were first bound to 25 µL of glutathione-coupled beads. The beads were then incubated with mock or the indicated amounts of synthetic RNA-5′-P or -5′-Pme in 0.5 mL of binding buffer 1 (50 mM Tris-HCl, pH 8, 50 mM NaCl and 0.5% NP-40) for 2 h at 4 °C. The beads were washed three times with 1 mL of binding buffer for 5 min at 4 °C. After the last wash, the beads were split into two tubes: half was used to check protein binding as above, and half was used to extract RNA with the Qiagen RNA Clean-up purification kit as in (Xhemalce et al, 2012) and eluted with 30 µL of water. Overall, 15 µL was used for analysis on Urea-PAGE gel. The gels were stained with SYBR-Gold prior to scanning.

The in vitro RNA binding assays in Fig. 2E,F, were done as in Fig. 2D until the last wash, after which the beads were resuspended in 50 µL of binding buffer. 25 µL were transferred to a 0.2-mL PCR tube, and irradiated with $UV_{254}$ at 400 mJ/cm². The beads were centrifuged for 1 min at 1000× g at 4 °C and the supernatant was removed. The GST protein was eluted with 25 µL of elution buffer (Glutathione 50 mM, in 50 mM Tris, HCl, 150 mM NaCl, final pH 8). 10 µL were separated on two NuPAGE™Novex™ 4–12% Bis-Tris gels, one of which was used for Silver Stain to visualize the proteins, and the other was transferred onto a PVDF membrane in 1× Novex™ Transfer buffer with 15% Methanol and 0.02% SDS. The membrane was air-dried for 1 h, reactivated with methanol, washed with 50 mL of 2× SSC/0.1% SDS for 2 min, and prehybridized with 10 mL of ULTRAhyb-Oligo Hybridization Buffer for 30 min at 37 °C in a roller bottle. The labeled probe was added directly to the ULTRAhyb-Oligo solution used for the prehybridization and hybridized for 18 h at 37 °C. Following hybridization, the membrane was transferred onto a box and washed twice with 50 mL of 2× SSC/0.5% SDS at 37 °C for 30 min with gentle agitation. The membrane was wrapped in a plastic reaction folder, sealed, and exposed on a phosphorimager screen.

## Silver stain

Protein or RNA samples were separated on a NuPAGE™ Novex™ 4–12% Bis-Tris gel or 15% Urea-PAGE gel, respectively, and stained using the FASTsilver Gel Staining Kit (#341298).

## Mass spectrometry

### Data collection
In total, 15 µL of each "heavy" and "light" pulldowns were mixed, and run on a 4–12% Bis-Tris gel. The gel was stained with Colloidal Coomassie and cut into ten pieces as shown on Fig. 1B. The gel pieces were digested with trypsin. The resulting peptides were cleaned with a C18 tip. Liquid chromatography was performed with a EASY-nLC™ 1000 Integrated Ultra High-Pressure Nano-HPLC System; a 15 cm long, 75 µM diameter C18 column (#164769) and MS/MS with a Q-EXACTIVE System equipped with a Nanospray Flex Ion Source as previously described (Abell et al, 2017).

### Data analysis
Raw spectra (ThermoFisher .RAW files) were processed directly using the MaxQuant software suite, version 1.5.1.2. Spectra were matched against the UniProt human protein database using the Andromeda search engine, followed by false discovery rate estimation and match filtering using a target-decoy approach. Protein identifications, peptide counts, heavy/light (H/L) ratios, and heavy and light intensities were reported. Using custom R scripts, normalized (zero-centered) paired forward and reverse H/L ratios were extracted from MaxQuant, log2 transformed, and plotted. Proteins showing a "true positive" pattern—for example, a positive log2 (H/L) ratio in the forward experiment and a negative log2 (H/L) ratio in the reverse experiment—relative to median log2 (H/L) values were identified.

## Recombinant protein purification

### Recombinant BCDIN3D protein
Full-length human BCDIN3D (residues 1–292) was cloned into a pFBOH-MHL donor plasmid, which is a derivative of the pFBOH-LIC Vector (GenBank accession EF456740). Production was done in Sf9 insect cells grown in HyQ® SFX medium (Fisher Scientific) infected with recombinant viral stock of BCDIN3D constructs. Harvested cell pellets were resuspended in a lysis buffer containing 50 mM Tris (pH 8.0), 500 mM NaCl, 5% glycerol, 0.5% NP-40, 5 mM TCEP and Benzonase (25 U/mL). Cobalt-charged TALON resin (Clontech) was used to capture N-terminally His₆-tagged BCDIN3D protein, followed by two washing steps with 20 CV of lysis buffer supplemented with 5 mM imidazole, and then with 0.5% sodium deoxycholate. BCDIN3D was additionally incubated with TEV at 1:50 ratio in a dialysis buffer containing 20 mM Tris (pH 8.0), 300 mM NaCl, and 2 mM TCEP O/N to cleave the tag. Proteins were further purified by size-exclusion chromatography Superdex200 (GE Healthcare Life Sciences) column, pre-equilibrated with 20 mM Tris (pH 8.0), 150 mM NaCl, 2 mM TCEP. Collected fractions were pooled together and concentrated up to 7 mg/mL and stored at −80 °C.

### Recombinant GST fusion proteins
The cDNAs of the 11 multisynthetase subunits were cloned into the pGEX-2TK-P plasmid in frame with a N-terminal GST tag. BL21 (DE3) pRIL E. coli cells carrying the plasmids were grown in 1 L of LB medium to OD₆₀₀ = 0.4 at 37 °C, and the recombinant protein expression was induced with 0.1 mM IPTG overnight at 18 °C. The cell pellet from the equivalent of 1000 OD was resuspended in 22.5 mL of pre-chilled PBS containing one tablet of EDTA-free Protease Inhibitor Cocktail from Roche. Upon transfer into a 50-mL tube, the cell suspension was sonicated with 3 × 30 s pulses at 30% amplitude using the tapered probe. In all, 2.5 mL of PBS-20% Triton-X was added to the lysate, which was then rotated for 1 h at 4 °C prior to centrifugation at 20,000× g at 4 °C. The cleared lysate was incubated with 1 mL of pre-washed Glutathione Sepharose High-Performance Beads (GE # 17-5279-01) for 1 h at 4 °C. After transfer into a Biorad column, the beads were washed once with 5 mL of PBS-1% Triton-X, twice with 5 mL of PBS, and once with 5 mL of 50 mM Tris, HCl pH 8, 150 mM NaCl, prior to triple elution with 0.5 mL of GSH 50 mM, in 50 mM Tris, HCl, 150 mM NaCl, final pH 8. Upon transfer into a 10 KDa dialysis cup, the elution solution was dialyzed for 1 h, then O/N, in 1 L of cold 50 mM Tris, HCl pH 8, 150 mM NaCl at 4 °C. After dialysis, glycerol was added to 10%. Proteins were run on a denaturing polyacrylamide gel and quantified by Coomassie staining analysis

alongside BSA standards [125 ng, 250 ng, 500 ng, 750 ng, 1 µg, 1.5 µg, and 2 µg]. The proteins were then aliquoted and stored at −30 °C.

IARS could not be expressed in sufficient amounts for purification from pGEX-2TK-P-IARS plasmid under any tested condition. Thus this protein was purchased as indicated below.

### Other proteins

EPRS-C-MYC/DDK (TP317559) and MARS-C-MYC/DDK (TP302932) were purchased from Origene. IARS-His-GST tag (MBS5304271) was purchased from MyBioSource.

## Co-immunoprecipitation (Co-IP)

For the experiments in Fig. 3A,C,E,G, $2 \times 10^7$ HeLa-S3-FlpIn-control and derivative cells grown at a density of $4–6 \times 10^5$ cells per mL were used per Co-IP. The cells were washed twice with 25 mL of cold PBS, extracted with 0.6 mL of cold co-IP buffer (20 mM HEPES pH 7.5, 150 mM NaCl, 20% glycerol, 0.1% NP-040, 1 mM EDTA, 0.1 mM PMSF supplemented with EDTA-free Complete Protease Inhibitor cocktail from Roche) for 1 h, at 4 °C, and cleared by centrifugation for 10 min at 15,000×$g$ at 4 °C. The supernatant was incubated for 4 h with 40 µL of pre-washed anti FLAG M2 conjugated beads (Sigma) at 4 °C. The beads were washed three times with 0.6 mL of co-IP buffer, once with 0.6 mL of TBS, and eluted with 100 µL of TBS containing 150 ng/µL of 3×FLAG peptide for 30 min at 4 °C. The eluates were split into two tubes: half was used to check proteins, and half was used to extract RNA with the Qiagen RNA Clean-up purification kit as in ref. (Xhemalce et al, 2012) and eluted with 15 µL of water. For the treatments with RNase A, the co-IPs were performed in the presence of 1.5 µL at 20 µg/µL of RNase A solution. Experiments in Fig. 3D, 5 µg of anti-GFP and 5 µg of anti-EPRS antibody were incubated for 3 h at 4 °C with rotation prior to the addition of the equivalent of 25 µL of Dynabeads Protein G (Life Technologies # 10003D) for another hour. After the washes, the beads were resuspended with 25 µL of 1× Laemmli Buffer and heated for 5 min at 95 °C.

## iCLIP-seq

### Cell collection and iCLIP-seq protocol

In total, $2.5 \times 10^7$ HeLa-S3-FlpIn control and BCDIN3D-KO cells grown in spinner flasks were collected by centrifugation at 200×$g$ for 5 min at 4 °C, and washed once with 25 mL of cold 1× PBS. The pellet was resuspended in 3 mL of cold 1×PBS and transferred on a 10-cm diameter plate placed on a cooling block. The plate was irradiated with $UV_{254}$ at 150 mJ/cm². The cells were transferred into a 50-mL Falcon tube, and the remaining cells were taken up with another 9.5 mL of cold PBS. The cell suspension was mixed well by pipetting and aliquoted into 12 1.5-mL tubes (1 mL aliquots at $2 \times 10^6$ cells). The tubes were centrifuged at 200×$g$ for 5 min at 4 °C, the supernatant was carefully aspirated, and the cell pellets were snap-frozen in liquid Nitrogen, and stored at −80 °C. The rest of the iCLIP-seq protocol was as previously described (Huppertz et al, 2014).

### iCLIP-seq data analysis

Raw data were demultiplexed with umi_tools, requiring a perfect index match to account for random bases adjacent to the index (Smith et al, 2017). After demultiplexing, raw reads were aligned

against a custom small RNA reference using STAR with default parameters except for the following modifications: [--outFilter-MultimapNmax 999999 --outFilterMultimapScoreRange 0 --out-SAMprimaryFlag AllBestScore --seedSearchStartLmax 15 --outFilterScoreMinOverLread 0.25 --seedSearchLmax 15] (Dobin et al, 2013). The custom small RNA reference was assembled by extracting raw sequences from several human genetic databases, adding 10 N of padding on the 5′- and 3′-end of each sequence, and concatenating into an alignment reference. These databases were: miRbase (microRNA (Kozomara et al, 2019)), piRNAdb (piwi RNAs (Piuco R 2021), tRNAdb (transfer RNAs (Jühling et al, 2009)), and mitotRNAdb (mitochondrial transfer RNAs (Jühling et al, 2009)), and a curated set of representative sequences from RefSeq (ribosomal RNA (Jühling et al, 2009)). Counts per small RNA were computed using samtools idxstats, and the resulting count matrices were visualized using custom scripts in R (O'Leary et al, 2016). In parallel, reads were also aligned against the human genome version hg38 using STAR with identical parameters except for the reference index. Uniquely aligned reads were quantified against Gencode v21 using featureCounts with default parameters (Liao et al, 2014).

## Cross-linking RNA immunoprecipitation (X-RIP)

Overall, $2 \times 10^7$ HeLa-S3-FlpIn control and BCDIN3D-KO cells grown in spinner flasks were cross-linked with 1% formaldehyde with gentle rotation at room temperature for 10 min. The cross-linking was stopped with 125 mM glycine with gentle rotation at room temperature for 5 min. The cells were collected by centrifugation for 5 min at 200×$g$ in a cold centrifuge. After washing twice with 25 mL of cold PBS, the cell pellets were frozen at −80 °C. After thawing for 10 min on ice, the cell pellets were resuspended in 1.6 mL of cold RIPA Buffer (50 mM Tris-HCl pH 7.5, 150 mM NaCl, 1 mM EDTA pH 8, 1% NP-40, 0.5% Sodium Deoxycholate, 0.05% SDS, freshly supplemented with Protease Inhibitors Cocktail—1 miniComplete EDTA-free Protease Inhibitors Cocktail tablet from Roche for 10.5 mL). The lysate was transferred into a polystyrene 15-mL conical tube and sonicated with the aid of probes with the Bioruptor for 10 min at high power 10 s on/10 s off at 4 °C temperature. After sonication, the lysate was transferred to a 2 mL tube and centrifuged for 10 min at 15,000×$g$ at 4 °C. The supernatant was transferred to a new 2-mL tube. In total, 440 µL of the supernatant was diluted with 660 µL of RIPA buffer. 100 µL was kept as input, and 1 mL was used for each immunoprecipitation in a 1.5-mL tube. 2.5 µg of anti-GFP or anti-EPRS antibody was added to each immunoprecipitation and incubated overnight (~18 h) with rotation at 4 °C. The next day, 25 µL of washed Protein G Dynabeads were added to each immunoprecipitation and incubated for 2 h with rotation at 4 °C. The beads were washed five times with 0.5 mL of High Stringency RIPA buffer (50 mM Tris-HCl, pH 7.5, 1 M NaCl, 1 mM EDTA, 1% NP-40, 1% Sodium Deoxycholate, 0.1% SDS, 1 M urea, 0.2 mM phenylmethyl-sulfonyl fluoride) at 4 °C. For each wash, the tube was centrifuged for 3 s at 1100×$g$ at 4 °C; placed on a magnetic rack on ice for 30 s; the supernatant was carefully removed; 0.5 mL of High Stringency RIPA buffer was added and nutated for 1 min at 4 °C. After the final wash, the immunoprecipitations were reverse-cross-linked and eluted by adding 100 µL of Elution Buffer (50 mM Tris–Cl, pH 7.0, 5 mM EDTA, 10 mM dithiothreitol (DTT), 1% SDS) to the protein G beads and incubating the tubes for 1 h at 70 °C at 500 rpm. The tubes were

centrifuged for 3 s at 1100× g, placed on a magnetic rack for 30 s, and the supernatant was transferred into a new 1.5-mL tube. The RNA was extracted from the supernatants with the Qiagen RNA Clean-up purification kit as in (Xhemalce et al, 2012) and eluted with three times 18 μL of water to obtain a total recovered volume of 51 μL. The inputs were similarly processed concomitantly. In total, 8 μL of RNA from input and each immunoprecipitation was used for reverse transcription with 2 μL of the QuantaBio qScript cDNA SuperMix (95048-25) according to the manufacturer's instructions. After reverse transcription, the solution was diluted with 40 μL of water and 2.5 μL of the diluted cDNA solution was used for real-time PCR.

## Polysome fractionation

In total, $2 \times 10^7$ HeLa-S3-FlpIn control and BCDIN3D-KO cells grown in spinner flasks were transferred into a 50 mL Falcon tube containing a volume of cycloheximide calculated to result in a final concentration of 100 μg/mL. The cells were centrifuged for 5 min at 200×g at 4 °C with the swing rotor of an Eppendorf 5810R centrifuge. The cells were washed twice with 25 mL of cold PBS with 100 μg/mL of cycloheximide. The cells were resuspended in 400 μL of polysome fractionation lysis buffer (20 mM Tris·HCl (pH 7.4), 150 mM NaCl, 5 mM MgCl$_2$, 1 mM DTT, 1% Triton X-100, 100 μg/mL cycloheximide), transferred into a 1.5 mL Eppendorf tube and lysed for 1 h at 4 °C on a nutator. The cells were centrifuged for 10 min at 15,000×g at 4 °C. For MDA-MB-231 cells, cells from one 254 × 254 mm square plate were lysed as in steps 1–6 in the procedure described in the Nature Protocol article from (Ingolia et al, 2012) using the flash freezing option without cycloheximide or harringtonine pre-treatment. For both cell types, the supernatant was layered on top of a linear 7–50% (w/v) sucrose gradient containing 100 μg/mL of cycloheximide. The tubes were centrifuged in a Beckman SW41TI rotor at 36,000 rpm for 2 h and 30 min at 4 °C. Polysome profiles were monitored by absorbance at 254 nm, and gradient fractions were collected on an ISCO density gradient fractionator. Proteins from each fraction were analyzed by Western Blot as indicated. RNA from each fraction was purified using the Qiagen RNeasy MinElute Cleanup Kit with a modified protocol that allows the recovery of RNAs of all sizes, as in ref. (Xhemalce et al, 2012). Briefly, 100 μL from each fraction was mixed with 350 μL of RLT buffer and 675 μL of 100% Molecular grade Ethanol. The mixture was passed through the Qiagen RNeasy MinElute column. The column was successively washed with 500 μL of RPE buffer and 750 μL of 80% Ethanol, dried by centrifugation, and the RNA was eluted with 30 μL of water. In all, 1 μL of RNA was analyzed on an Agilent Total Eukaryotic RNA Pico Chip and 8 μL was used for reverse transcription with 2 μL of the QuantaBio qScript cDNA SuperMix (95048-25) according to the manufacturer's instructions. After reverse transcription, the solution was diluted with 40 μL of water and 2.5 μL of the diluted cDNA solution was used for real-time PCR.

## Ribo-seq

Cells from one 100-mm diameter plate were used to perform ribosome profiling as in (Ingolia et al, 2012) using the flash freezing option with harringtonine pre-treatment. Bioinformatic analysis was performed as in (Reinsborough et al, 2021).

## Cellular fractionation

In total, $10^6$ HeLa-S3-FlpIn cells grown in spinner flasks were collected by centrifugation at 200×g for 5 min at 4 °C, and washed twice with 5 mL of cold PBS. The cells were then fractionated with the Subcellular Protein Fractionation Kit for Cultured Cells from Thermo Scientific (#78840) according to the manufacturer's instructions.

## Mitochondrial enrichment

Overall, $2 \times 10^7$ HeLa-S3-FlpIn control and BCDIN3D-KO cells grown in spinner flasks were collected by centrifugation at 200×g for 5 min at 4 °C, and washed once with 25 mL of cold PBS. The cells were then fractionated with the Mitochondria Isolation Kit for Cultured Cells from Thermo Scientific (#89874) according to the manufacturer's instructions with Option A (Isolation of Mitochondria using Reagent-based Method). Per the recommendation of the manufacturer, at step 7 of the protocol, the lysate was centrifuged at 3000×g for 15 min instead of 12,000×g to obtain a more purified fraction of mitochondria, with >50% reduction of lysosomal and peroxisomal contaminants. The supernatant from $2 \times 10^7$ cells was in a final volume of 2 mL, while the mitochondria-enriched fraction was in a final volume of 50 μL. In all, 10 μL of each fraction was analyzed by western blot.

## Western blot

Proteins were separated in SDS-PAGE gel and transferred onto a 0.45 μM PVDF or Nitrocellulose membrane in 1× Towbin Buffer with 15% or 20% Methanol and 0.02% SDS for 90 min at 400 mA. The membranes were blocked for 30 min at room temperature in TBS-TM (Tris-buffered saline, 0.1% Tween 20, 5% Nonfat Dry Milk from Cell Signaling #9999) and incubated overnight at 4 °C with TBS-TM buffer containing the indicated antibodies. The membranes were washed three times 10 min with TBS-T, incubated 1 h with TBS-T containing the appropriate secondary antibodies, washed, and revealed with ECL (Amersham). The ECL signal was detected with either the Syngene 05-GBOX-CHEMI-XR5 or by exposure to film. For LI-COR western blots, Immobilon®-FL membrane was used in combination with Odyssey Blocking Buffer (TBS) and IRDye 680RD Goat anti-Rabbit and IRDye 80CW Goat anti-Mouse secondary antibodies. The blots were scanned using an Odyssey imaging system and quantified with ImageJ. The list of all primary antibodies is in Appendix Table S3.

## RNA analysis

For reverse transcription, 500 ng of total RNA or 8 μL of RNA from various applications was denatured for 5 min at 65 °C, followed by immediate cooling on ice. 2 μL of QuantaBio qScript cDNA SuperMix (95048-25) was added to the 8 μL and was incubated in a PCR machine for 5 min at 25 °C, 40 min at 42 °C, and 5 min at 80 °C. Subsequently, 40 μl was used to dilute the cDNA and 2.5 μL was used for real-time PCR with gene-specific primers on a StepOne Plus system. The list of all oligonucleotides is in Appendix Table S4.

The northern blots were performed as previously described (Xhemalce and Kouzarides, 2010).

For 5′-Pme analysis, 1 μg of total RNA was treated with Mock or Tobacco Acid Pyrophosphatase (TAP) from Epicenter (T81050) in a total volume of 20 μL in 1×TAP buffer (50 mM sodium acetate (pH6), 1 mM EDTA, 0.1% β-mercaptoethanol, 0.01% Triton) and 1 μL of water (Mock) or 1 μL of TAP at 5U/μL for 1 h at 37 °C. Each reaction's volume was brought up to 100 μL with water prior to RNA purification with the Qiagen RNA Clean-up purification kit as in ref. (Xhemalce et al, 2012) and elution with 17 μL of water. In total, 7 μL of each resulting RNA was treated with Mock or Terminator from Epicenter (TER51020) in a total volume of 20 μL in 1× Terminator Reaction A buffer with 0.5 μL of of RNaseOUT at 40 U/μL from Invitrogen and 1 μL of water (Mock) or 1 μL of Terminator at 1 U/μL for 2 h at 30 °C. Each reaction's volume was brought up to 100 μL with water prior to RNA purification with the Qiagen RNA Clean-up purification kit as in ref. (Xhemalce et al, 2012) and elution with 17 μL of water. 1 μL of each RNA was analyzed in a Bioanalyzer Pico total RNA chip to check for successful digestion of the 28 S, 18 S and 5.8 S 5′-P rRNAs by Terminator; 7 μL of each RNA was analyzed in a denaturing 15% Urea polyacrylamide gel to check for successful digestion of 7SK, U4, and other 5′-PPP or TMG snRNAs only by dual TAP+Terminator treatment as well as to check for equal RNA recovery in all samples by inspection of the mature tRNAs which are resistant to Terminator treatment (Devanathan et al, 2021; Reinsborough et al, 2019); and the remaining 8 μL was used for reverse transcription with 2 μL of the QuantaBio qScript cDNA SuperMix (95048-25) according to the manufacturer's instructions. After reverse transcription, the solution was diluted with 40 μL of water and 2.5 μL of the diluted cDNA solution was used for real-time PCR with the PowerUp SYBR mix and specific primers listed in Appendix Table S4. The dual TAP+Terminator-resistant fraction of B2M, LRPPRC and PGK1 mRNAs was calculated by the ratio of (TAP+Terminator)/(TAP+Mock), which is equal to $2^{\wedge}(Ct^{TAP+Mock} - Ct^{TAP+Terminator})$.

### Flow cytometry

HeLa-S3-FlpIn and HeLa-S3-FlpIn-LRPPRC-5′UTR-GFP Control and BCDIN3D-KO cells were grown in flasks and 1 mL of cells were directly analyzed in a NovoCyte benchtop flow cytometer with the following parameters: laser 488 nm; filter 530 ± 30 nm; standard debris and doublet exclusion; ∼ 30,000 single cells. FCS files were analyzed with FlowJo and Floreada.

### Live-cell analysis and Mitotracker treatment

HeLa-FlpIn-LRPPRC-5′UTR-GFP cells were reverse transfected with 10 nM siRNAs control (siNC) and against BCDIN3D (siBCDIN3D) in 36-mm diameter dishes using Lipofectamine RNAiMAX and images were taken with a Leica fluorescent DM II LED microscope 50 h after transfection. To stain mitochondria, cells were washed once with 4 mL of pre-warmed PBS, then incubated for 15 min with 4 mL of pre-warmed PBS containing 100 nM MitoTracker-Red CMXRos. The cells were washed again with 4 mL of pre-warmed PBS and imaged again with the Leica fluorescent microscope. Images were analyzed on ImageJ.

## Data availability

This study includes no data deposited in external repositories. All data needed to evaluate the conclusions in the paper are present in the article, the Appendix, or the associated source data.

## Peer review information

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

## Acknowledgements

This research in the Xhemalçe lab was supported by the Welch Foundation (Grant number: F1859), the Department Of Defense—Congressionally Directed Medical Research Program—Breast Cancer Breakthrough Award (W81XWH-16-1-0352), NIH Grant R01 GM127802, a grant from STORM Therapeutics and

start-up funds from the Institute of Cellular and Molecular Biology and the College of Natural Sciences at the University of Texas at Austin, USA. The Structural Genomics Consortium is a registered charity (number 1097737) that receives funds from AbbVie, Bayer Pharma AG, Boehringer Ingelheim, Canada Foundation for Innovation, Eshelman Institute for Innovation, Genome Canada through Ontario Genomics Institute [OGI-055], Innovative Medicines Initiative (EU/EFPIA) [ULTRA-DD grant no. 115766], Janssen, Merck KGaA, Darmstadt, Germany, MSD, Novartis Pharma AG, Ontario Ministry of Research, Innovation and Science (MRIS), Pfizer, São Paulo Research Foundation-FAPESP, Takeda, and Wellcome. We would like to thank Dr. Matthias Hentze (EMBL) and Dr. Margaret Price (University of Iowa) for the gracious sharing of the HeLa-FlpIn-Trex cell line; Dr. Shelley Payne (University of Texas at Austin) for the use of her lab's ultracentrifuge, Dr. Paul Macdonald (University of Texas at Austin) for the use of his lab's fraction collector, Dr. Larissa Durfee in the Huibregtse lab (University of Texas at Austin) for technical advice on polysome fractionation, Dr. Kyle M Miller (University of Texas at Austin) for the use of his confocal microscope, and Dr. Lulu Cambronne (University of Texas at Austin) for the use of her lab's flow cytometer.

## Author contributions

**Hélène Ipas**: Validation; Investigation; Methodology; Writing—review and editing. **Ellen B Gouws**: Validation; Investigation; Methodology; Writing—review and editing. **Nathan S Abell**: Formal analysis; Investigation; Methodology; Writing—review and editing. **Po-Chin Chiou**: Investigation; Methodology; Writing—review and editing. **Sravan K Devanathan**: Investigation; Methodology. **Solène Hervé**: Investigation; Methodology. **Sidae Lee**: Investigation; Methodology. **Marvin Mercado**: Investigation; Methodology. **Calder W Reinsborough**: Investigation; Methodology. **Levon Halabelian**: Investigation; Methodology. **Cheryl H Arrowsmith**: Supervision; Funding acquisition. **Blerta Xhemalçe**: Conceptualization; Resources; Data curation; Software; Formal analysis; Supervision; Funding acquisition; Validation; Investigation; Visualization; Methodology; Writing—original draft; Project administration; Writing—review and editing.

## Disclosure and competing interests statement

The authors declare no competing interests.

