## [Peer Review File · EMBO Reports]

ChemRAP uncovers specific mRNA translation regulation via RNA 5'-Pme

Hélène Ipas, Ellen Gouws, Nathan Abell, Po-Chin Chiou, Sravan Devanathan, Solène Hervé, Sidae Lee, Marvin Mercado, Calder Reinsborough, Levon Halabelian, Cheryl Arrowsmith, and Blerta Xhemalce

DOI: [10.15252/embr.202358555](https://doi.org/10.15252/embr.202358555)

Corresponding author(s): Blerta Xhemalce (b.xhemalce@austin.utexas.edu)

Review Timeline:

Transfer Date:	27th Nov 23
Editorial Decision:	13th Dec 23
Revision Received:	20th Dec 23
Accepted:	3rd Jan 24

Editor: Esther Schnapp

Transaction Report: A revised version of this manuscript was transferred to EMBO reports following peer review at the EMBO Journal.

Date: 17th Mar 23 09:07:27
Last Sent: 17th Mar 23 09:07:27
Triggered By: Kelly Anderson
From: k.anderson@embojournal.org
To: b.xhemalce@austin.utexas.edu
Subject: Manuscript EMBOJ-2023-113533 - Decision
Message: Dear Dr. Xhemalce,

Thank you for submitting your manuscript for consideration by the EMBO Journal. It has now been seen by three referees whose comments are shown below.

Unfortunately, all three referees raise major concerns (including regarding the methods, interpretation, and biological significance) that we do not think would be addressable in the standard revision time. Given these negative opinions and the fact that the EMBO Journal can only afford to accept papers which receive enthusiastic support from a majority of referees, I am afraid we cannot offer to publish it here.

Thank you in any case for the opportunity to consider this manuscript. I am sorry we cannot be more positive on this occasion, but we hope nevertheless that you will find our referees' comments helpful.

Yours sincerely,

Kelly M Anderson, PhD
Editor
The EMBO Journal
k.anderson@embojournal.org

Referee #1:

In their manuscript, Ipas and colleagues identify proteins binding to the RNA modification 5'Pme mediated by BCDIN3D. BCDIN3D monomethylates the 5'-monophosphate of cytoplasmic tRNA^{His} and precursor miRNA. 5'Pme in synthetic chemically modified microRNA duplexes was bound by EPRS, a multifunctional aminoacyl-tRNA synthetase that catalyzes the aminoacylation of glutamic acid and proline tRNA species. Binding interaction of BCDIN3D with EPRS regulated translation and subcellular localization of LRPPRC, a multifunctional mitochondrial protein.

To fully understand the molecular and cellular functions of RNA modifications, it is important to identify the respective modification-specific reader proteins.

However, my main concern with this manuscript is that it is full of individual experiments that do not add up to a coherent story. The authors demonstrate that EPRS recognizes 5'Pme on a synthetic RNA construct, but do not confirm these data with tRNA^{His}, another known 5'Pme-modified RNA. Pull-down and iCLIP experiments are not validated with a second independent method and are not consistent with expected results from the literature. But then the authors use the tRNA^{His} sequence and structure as an argument for BCDIN3D methylation and EPSR-binding in LRPPRC mRNA. They conclude that 5'Pme sites in LRPPRC regulate translation and subcellular localization. It is entirely unclear to me what the physiological function these findings could have, and no model or hypothesis is provided.

Specific comments:

1. The pull-down experiment uses chemically synthesized RNAs to identify 5'Pme binding proteins. One of the best described BCDIN3D-modified RNA seems to be cytoplasmic tRNA^{His}. The authors need to confirm their pull-down experiments with this second validated target RNA. This seems particularly important given that the authors identify a multifunctional aminoacyl-tRNA synthetase which should not (or is not expected to) bind pre-miRNAs or mRNAs.
2. The literature describes BCDIN3D to specifically recognize a unique feature only found in tRNA^{His}. It recognizes the acceptor helix of tRNA^{His} with a G-1:A73 mispair at the top of the eight-nucleotide-long acceptor helix and the G-1 nucleobase (Martinez et al., 2017). Does the synthetic RNA form a similar structure? How does this relate to LRPPRC?
3. Line 193: the authors state that they cannot measure significant differences in binding of EPSR to the cognate tRNA^{Glu} aminoacylated by its ERS domain between BCDIN3D control and knockout cells and conclude that BCDIN3D may not regulate EPSR's canonical functions. It is unclear to me why it would bind when the 5'Pme is very specifically added to tRNA^{His}. It seems no conclusion can be drawn from the iCLIP experiments in particular when no positive control (miRNAs or tRNA^{His}) was present in the sequencing data.
4. The authors instead find some mRNAs bound to EPRS in control cells that are enriched in codons decoded by tRNAs charged by the tRNA synthetases of the multi-synthetase complex. The physiological relevance of this finding is entirely unclear to me. Do the authors suggest that the mRNA is aminoacylated? Alternatively, it has been shown that mRNA association by aminoacyl tRNA synthetases can occur at a putative anticodon mimics and autoregulate translation in response to tRNA levels. But the authors do not show any evidence for that.
5. The authors then move on to show that BCDIN3D caps the 5' end of LRPPRC mRNA. The relevance of the experiments with respect to EPSR is also unclear to me.
6. The authors do not directly show that BCDIN3D methylates LRPPRC mRNA or that this would directly alter EPSR binding.
7. Little conclusion can be drawn from the mitochondrial localization studies as no classical mitochondrial marker was included as positive control. In addition, no evidence is provided that mitochondria function and number are unaffected by BCDIN3D knockout.

Minor comments:

1. The official gene name for EPRS in human cells is EPRS1.

Martinez, A., Yamashita, S., Nagaike, T., Sakaguchi, Y., Suzuki, T., and Tomita, K. (2017). Human BCDIN3D monomethylates cytoplasmic histidine transfer RNA. *Nucleic Acids Res* 45, 5423-5436. 10.1093/nar/gkx051.

Referee #2:

In their manuscript "ChemRAP uncovers specific mRNA translation regulation by the multisynthetase complex via RNA 5'-Pme" Ipas et al. utilize ChemRAP as high-throughput method to unbiasedly identify RNA binding proteins that specifically interact with 5'-Pme-modified RNA. The authors further suggest that 5'-Pme RNA modification mediates the interaction between BCDIN3D, EPRS and the multisynthetase complex to regulate the expression of specific mRNAs.

The manuscript is nicely written and the presentation is mostly clear, nevertheless, I have some concerns about the interpretation of the results and their real significance, since the conclusions are based on very subtle differences, where the molecular and biological significance remains to be proven. Therefore, I don't support the publication of this manuscript at this stage.

Major points:

1) The interaction with the multisynthetase complex is not surprising since tRNAs are 5'-Pme modified. Nevertheless, the authors could only confirm in vitro a slight preference for EPRS to bind 5'-Pme. In addition, from the components of the multisynthetase complex that bind to the two RNA baits in Fig 2, the authors excluded AIMP1, DARS, LARS and KARS from the discussion, which prefer to bind to 5'-P, and AIMP2, QARS, RARS, which don't bind at all. All these proteins were among the highest enriched proteins in the ChemRAP experiment. Therefore, considering the really slight preference of EPRS for the 5'-Pme bait, the in vitro pull-down (Fig 2C) doesn't validate the ChemRAP (Fig 2B).

In addition, the sentence in row 118: "EPRS binds directly to the 5'-Pme modification on the RNA, while ..." should be rephrased, since the authors observe only a slight preference of EPRS for 5'-Pme, but EPRS binds both to the 5'-Pme and 5'-P baits.

A competitive binding assay to show the preference of EPRS for 5'-Pme over 5'-P is missing here.

2) Row 137: It is not clear how the interaction between BCDIN3Df, MARS and EPRS could "independently support" the ChemRAP findings, since all the other

multisynthetase complex members identified in the ChemRAP experiment are not tested (or not shown) here.

3) Fig 4A. The authors write: "This image clearly shows that EPRS, which migrates at 187 ~170 KDa, interacts with RNA. Moreover, BCDIN3D-KO severely decreases the levels of RNAs directly interacting with EPRS." Here a quantification of RNA binding in the two genotypes it is needed.

4) The fact that α EPRS iCLIP-seq could not detect or identify changes in the known and much more abundant BCDIN3D RNA targets, like tRNAs and miRNAs, raises doubts on the specificity of the assay. Therefore, a validation of the results is needed. A possibility is that tRNAs and miRNAs could not be efficiently sequenced. α EPRS iCLIP-Northern with probes specific to cognate tRNAs or known miRNAs could help.

5) Figure 4D-E: Again, a validation with known BCDIN3D targets is needed.

6) Row 269: To show that the 5'UTR of LRPPRC mRNA is sufficient for "more translation" of the GFP reporter in BCDIN3D-KO cells, polysome profiling experiments are necessary, since the approach with GFP reporters used by the authors only shows that there is more protein, and not that the mRNA is translated more efficiently.

7) Figure 4K-L and row 300: The hypothesis that the translation of LRPPRC requires the localization of its mRNA near mitochondria for proper mitochondrial transport is intriguing, but the evidence is based on very slight differences, that would need further validation in an independent cell line, or best a relevant more physiological model system (rather than cancer cells).

In addition, clear prove, like the localization of the 5'-Pme modified fraction of LRPPRC mRNA next to the mitochondria or other membranes, is missing here. Moreover, if 5'-Pme acts as localization or translation signal of the RNA near mitochondria, how can this function be explained for the other known 5'-Pme modified RNAs, like tRNA and miRNA. To my knowledge, no enrichment of tRNA^{His} near mitochondria has been reported.

Minor points:

1) In Fig1b, H and L are swapped respect to Fig 1A. This is misleading, please clarify.

2) Figure 4 is really crowded. Better organization or splitting in two is needed.

3) In row 136 the authors write: "BCDIN3Df interacts in cells with MARS and EPRS and other members of the multisynthetase complex (Fig 3A and data not shown)." I think these data should be shown or not mentioned here.

Referee #3:

Review of Ipas et al.: "ChemRAP uncovers specific mRNA translation regulation by the multisynthetase complex via RNA 5' phospho-methylation"

This study uses RNA pull-down and Mass Spectroscopy to identify protein binders to 5' methylated monophosphorylated RNA(5'Pme-RNA). The multisynthetase complex (MSC) is identified and the interaction with the complex is found to depend on the EPRS subunit. Furthermore, evidence is presented in favour of EPRS binding to specific mRNAs in a 5'Pme/BCDIN3D dependent

manner with the LRPPRC mRNA-EPRS interaction being characterised further. These are interesting findings and the possibility that 5'Pme should regulate translation of specific mRNAs via recruitment of MSC could be an important and surprising discovery. The manuscript demonstrates that EPRS interacts with 5'Pme and this is the best characterised and most important finding of the study. Less convincing is the part of the manuscript showing that EPRS interact with specific messenger RNAs depending on interaction with the 5'Pme writer BCDIN3D. However, the presented data does not fully support the claims of the manuscript and additional experimentation will be needed to ensure that the conclusions are valid. In the current form, I therefore cannot recommend the publication this manuscript.

Major points:

1. SILAC MS experiment replicates. The authors state that they have three biological replicates of the experiments presented in Fig. 1C and 1D in main text (L78). These should be used for statistical analysis of the data and used for showing volcano plots or similar.
2. The authors use CLIP-seq to identify the EPRS binding mRNAs but very few RNAs are detected, a positive control is missing and in combination this makes it difficult to interpret and trust the data. The EPRS linker has previously been shown to bind to RNA/the GAIT element (PMID 18374644, 22386318, 19647514 and more), these papers are not cited and discussed, and the GAIT elements are not detected in the analysis. The authors focus on the LRPPRC mRNA, which is one of the detected RNAs but based on the above the evidence for the interaction is not convincing. Phosphorylation of the EPRS linker may release EPRS from the MSC, allowing RNA interaction (with GAIT) and this seems to be of potential importance for the findings in this study, but is not analysed or discussed.
3. The study shows that BCDIN3D can 5'Pme modify a truncated version of the LRPPRC mRNA, which has an artificial dsRNA "blunt" end. However, in vivo the 3' end would extend further and in addition mRNAs are generally believed to be efficiently m7GpppN capped co-transcriptionally, meaning that a 5'Pme dependent mechanism of EPRS recruitment is hard to reconcile with normal mRNA processing. The methylation could occur after mRNA decapping but this remains speculative and clearly a demonstration that the LRPPRC mRNA (or any other mRNA) is 5'Pme modified to a biologically relevant level would strengthen the study a lot.
4. For the data in figure 3C, it would make sense to also show the non-MSC interactor to give some idea of the variation in the experiment.
5. I appreciate that the authors mostly refrain from overstating their result and use "suggest", "may" and "consistent with", however I still think that both the title, abstract and discussion make claims that is not supported by the data.

** As a service to authors, EMBO Press provides authors with the possibility to transfer a manuscript that one journal cannot offer to publish to another EMBO publication or the open access journal Life Science Alliance launched in partnership between EMBO Press, Rockefeller University Press and Cold Spring Harbor Laboratory Press. The full manuscript and if applicable, reviewers' reports, are automatically sent to the receiving journal to allow for fast handling and a prompt decision on your manuscript. For more details of this service, and to transfer your manuscript please click on

Link Unavailable

ANSWERS TO REVIEWERS

We would like to thank the reviewers for their time and constructive suggestions that helped us to strengthen our manuscript. Our point-by-point responses are included below, with the original questions and comments in bold, and our answers in regular text. We have also highlighted the changes in the main text of the manuscript in teal color.

The most important points that we have addressed in the revised manuscript are:

- ⇒ As requested by reviewer #3, and to clarify comments from reviewers #1 and #2, we now provide an expanded explanation of the non-canonical roles of EPRS mediated by its linker domain, which was previously shown to not bind tRNA but to bind hairpin structures within the 3'-UTR of ceruloplasmin and VEGF-A mRNAs as part of the interferon gamma (IFN- γ) activated inhibitor of translation (GAIT) system that dampens inflammation upon IFN- γ pathway stimulation in **lines 144-153** (see schematic below from Fig 4 of Arif *et al.* review on the GAIT system (PMID: 29152905)). Additionally, In the revised manuscript, we show that BCDIN3D binds mRNA 5'UTRs with 5'-Pme modification to control translation of specific mRNAs in the absence of IFN- γ stimulation. In support of this, we have added new LC-MS/MS data showing that EPRSt does not interact with GAIT complex components in control or BCDIN3D-KO cells in the **new Fig 3E** to answer comments by reviewer #3, as well as that EPRSt does not interact with tRNA^{His} using northern blotting in the **new Fig 3G** to respond comments by reviewers #1 and #2.
- ⇒ As requested by all reviewers, we have now strengthened our results showing that BCDIN3D methylates the LRPPRC mRNA by performing a dual treatment of RNA with \pm Tobacco acid pyrophosphatase (TAP) and \pm Terminator. TAP decaps mRNA to 5'-P and Terminator degrades RNAs with 5'-P ends, but not 5'-Pme ends. We show that 20 to 25% of the LRPPRC mRNA is resistant to this dual treatment in a BCDIN3D-dependent manner, strengthening our results that LRPPRC is 5'-Pme by BCDIN3D in cells. These new results are shown in the **new Fig 4E-F-G** and described in **lines 263-276**. We have also extended our results by including the PGK1 mRNA, which is also bound by EPRS at the start codon in many of our cellular analyses.
- ⇒ As requested by reviewer #2, we have performed polysome profiling that shows increased presence in the denser polysome fractions of LRPPRC mRNA in BCDIN3D-KO compared to control cells (**new Fig 5C** and **lines 292-305**). Additionally, we have performed ribosome footprinting coupled to next generation sequencing (Ribo-seq) in the presence of Harringtonine to specifically probe translation initiation. This Ribo-seq data shows increased initiation of translation of the LRPPRC and PGK1 mRNA in BCDIN3D-KO compared to control cells (**new Fig 5D** and **lines 292-305**). Together with the previous reporter assay data, our results support that BCDIN3D controls translation of specific mRNAs bound by EPRS around the start codon.

Of course, our paper does not address all of the interesting questions with respect to the molecular mechanisms and biological significance of the results shown here. As mentioned by reviewer #3, the GAIT system or readers of other RNA modifications required dozens of papers to begin having a clearer idea of their mechanism of action. Thus, we humbly ask to the reviewers to accept the limitations of our paper that we have extensively mentioned all along the manuscript. Nevertheless, our paper represents an important and significant first step towards understanding translation regulation by 5'-Pme and the multisynthetase complex.

Figure 4 from Arif *et al.* review

FIGURE 4 | Schematic diagram of glutamyl-prolyl tRNA synthetase (EPRS) domains, phospho-sites and their interactors. An N-terminal glutathione-S-transferase (GST)like domain is upstream of the ERS domain, which is joined to the FRS domain by a linker containing three WH2 repeats: R1, R2, and R3. In humans, NSAP1 binding to EPRS in the R2-R3 region requires Ser⁸⁸⁶ phosphorylation. Phospho-Ser⁷⁷-L13a and glyceraldehyde-3-phosphate dehydrogenase (GAPDH) interaction with EPRS requires Ser⁹⁹⁹ phosphorylation. The interferon γ -activated inhibitor of translation (GAIT) element RNA binds in the upstream R1-R2 repeat region.

REFEREE #1:

In their manuscript, Ipas and colleagues identify proteins binding to the RNA modification 5'Pme mediated by BCDIN3D. BCDIN3D monomethylates the 5'-monophosphate of cytoplasmic tRNA^{His} and precursor miRNA. 5'Pme in synthetic chemically modified microRNA duplexes was bound by EPRS, a multifunctional aminoacyl-tRNA synthetase that catalyzes the aminoacylation of glutamic acid and proline tRNA species. Binding interaction of BCDIN3D with EPRS regulated translation and subcellular localization of LRPPRC, a multifunctional mitochondrial protein.

To fully understand the molecular and cellular functions of RNA modifications, it is important to identify the respective modification-specific reader proteins. However, my main concern with this manuscript is that it is full of individual experiments that do not add up to a coherent story. The authors demonstrate that EPRS recognizes 5'Pme on a synthetic RNA construct, but do not confirm these data with tRNA^{His}, another known 5'Pme-modified RNA. Pull-down and iCLIP experiments are not validated with a second independent method and are not consistent with expected results from the literature. But then the authors use the tRNA^{His} sequence and structure as an argument for BCDIN3D methylation and EPRS-binding in LRPPRC mRNA. They conclude that 5'Pme sites in LRPPRC regulate translation and subcellular localization. It is entirely unclear to me what the physiological function these finding could have, and no model or hypothesis is provided.

Specific comments:

1. The pull-down experiment uses chemically synthesized RNAs to identify 5'Pme binding proteins. One of the best described BCDIN3D-modified RNA seems to be cytoplasmic tRNA^{His}. The authors need to confirm their pulldown experiments with this second validated target RNA. This seems particular important given that the authors identify a multifunctional aminoacyl-tRNA synthetase which should not (or is not expected to) bind pre-miRNAs or mRNAs.

- ⇒ We would like to explain that our goal in this paper was not to explore tRNA^{His} 5'Pme. The reason is that BCDIN3D has a high affinity for tRNA^{His}, and very low levels of BCDIN3D are sufficient to methylate 100% of tRNA^{His} in the cell. For example, triple negative breast cancer cells that are depleted for 70% of BCDIN3D compared to control cells, do not display any defect in tRNA^{His} 5'Pme, but do display profound effects on their transcriptome and tumorigenic phenotypes likely due to other regulatable BCDIN3D targets. We have now included this explanation in the introduction of the manuscript (lines 58-65). This phenomenon is not limited to BCDIN3D, but has been observed for other RNA modifiers such as the ac⁴C acetyltransferase NAT10. Furthermore, we now clearly show absence of binding of EPRS to tRNA^{His} by northern blot in the new Fig 3G.
- ⇒ EPRS has many non-canonical functions, mainly mediated by its linker domain. Previous work has shown that the EPRS linker domain does not bind to tRNAs, but does bind hairpin structures within the 3'-UTR of ceruloplasmin and VEGF-A mRNAs as part of the interferon gamma (IFN- γ) activated inhibitor of translation (GAIT) system that dampens inflammation upon IFN- γ pathway stimulation (Arif *et al*, 2018; Sampath *et al*, 2004b). Thus, it is not entirely unexpected that EPRS binds mRNA instead of tRNA through its linker domain. We have now included an expanded explanation of the GAIT system in the text of the manuscript in lines 144-153, immediately following the description of our finding that EPRS binds 5'Pme through its linker domain.

2. The literature describes BCDIN3D to specifically recognize a unique feature only found in tRNA^{His}. It recognizes the acceptor helix of tRNA^{His} with a G-1:A73 mismatch at the top of the eight-nucleotide-long acceptor helix and the G-1 nucleobase (Martinez et al., 2017). Does the synthetic RNA form a similar structure? How does this relate to LRPPRC?

- ⇒ As can be seen in Fig 4H, the 5' end of the LRPPRC mRNA is predicted to form an extended hairpin structure, with the first two nucleotides being unpaired and followed by an imperfect helix with 8 base pairs. This is similar to the tRNA^{His} structure and the experiment in Fig 4H shows that BCDIN3D needs this extended double stranded structure for 5'Pme activity. This is pointed out in the description of the results.

3. Line 193: the authors state that they cannot measure significant differences in binding of EPRS to the cognate tRNA^{Glu} aminoacylated by its ERS domain between BCDIN3D control and knockout cells and conclude that BCDIN3D may not regulate EPRS's canonical functions. It is unclear to me why it would bind when the 5'Pme is very specifically added to tRNA^{His}. It seems no conclusion can be drawn from the iCLIP experiments in particular when no positive control (miRNAs or tRNA^{His}) was present in the sequencing data.

⇒ The positive control of the iCLIP-seq experiments with the anti-EPRS antibody is tRNA^{Glu}, which is a tRNA charged by EPRS. As is the case with m⁶A RNA modification readers, we do not have an expectation that all BCDIN3D targets bind to EPRS. Other RNA sequence or structural elements, as well as competing RNA binding proteins are additional factors that may affect binding. For example, EPRS may not bind to tRNA^{His}, because tRNA^{His} has a very high affinity for BCDIN3D or other proteins.

4. The authors instead find some mRNAs bound to EPRS in control cells that are enriched in codons decoded by tRNAs charged by the tRNA synthetases of the multi-synthetase complex. The physiological relevance of this finding is entirely unclear to me. Do the authors suggest that the mRNA is aminoacylated? Alternatively, it has been shown that mRNA association by aminoacyl tRNA synthetases can occur at a putative anticodon mimics and autoregulate translation in response to tRNA levels. But the authors do not show any evidence for that.

⇒ We do not think that mRNA is aminoacylated. Additionally, the sequences obtained in the EPRS iCLIP-seq are not enriched in anti-codon mimics, so what we report in our manuscript is unrelated to the previous findings mentioned here. Although we cannot explain everything that we observe in this single paper, we find that it is interesting to explore why EPRS may be directly binding to mRNA in regions enriched with MSC codons in future work. It may be part of a system where the MSC needs to be in proximity of stretches of these codons to increase the availability of charged tRNAs locally.

5. The authors then move on to show that BCDIN3D caps the 5' end of LRPPRC mRNA. The relevance of the experiments with respect to EPRS is also unclear to me.

⇒ We hypothesized that BCDIN3D may methylate the LRPPRC mRNA because we found that EPRS binds to LRPPRC mRNA in control but not in BCDIN3D-KO cells.

6. The authors do not directly show that BCDIN3D methylates LRPPRC mRNA or that this would directly alter EPRS binding.

⇒ We have now strengthened our results showing that BCDIN3D methylates the LRPPRC mRNA by performing a dual treatment of RNA with ±Tobacco acid pyrophosphatase (TAP) and ± Terminator. TAP decaps mRNA to 5'-P and Terminator degrades RNAs with 5'-P ends, but not 5'-Pme ends. We show that 20 to 25% of the LRPPRC mRNA is resistant to this dual treatment in a BCDIN3D-dependent manner, strengthening our results that LRPPRC is 5'-Pme by BCDIN3D in cells. These new results are shown in the new Fig 4E-F-G and described in lines 263-276.

7. Little conclusion can be drawn from the mitochondrial localization studies as no classical mitochondrial marker was included as positive control. In addition, no evidence is provided that mitochondria function and number are unaffected by BCDIN3D knockout.

⇒ As suggested by the reviewer, we have now added a new mitochondrial marker, HSP60, as well as PGK1 that is both cytoplasmic and mitochondrial, to show that the results obtained with LRPPRC are not due to a global protein transport defect into mitochondria.

Minor comments:

1. The official gene name for EPRS in human cells is EPRS1.

⇒ We have now indicated the official name of EPRS in the paper (line 69).

Martinez, A., Yamashita, S., Nagaike, T., Sakaguchi, Y., Suzuki, T., and Tomita, K. (2017). Human BCDIN3D monomethylates cytoplasmic histidine transfer RNA. *Nucleic Acids Res* 45, 5423-5436. 10.1093/nar/gkx051.

REFEREE #2:

In their manuscript "ChemRAP uncovers specific mRNA translation regulation by the multisynthetase complex via RNA 5'-Pme" Ipas et al. utilize ChemRAP as high-throughput method to unbiasedly identify RNA binding proteins that specifically interact with 5'-Pme-modified RNA. The authors further suggest that 5'-Pme RNA modification mediates the interaction between BCDIN3D, EPRS and the multisynthetase complex to regulate the expression of specific mRNAs.

The manuscript is nicely written and the presentation is mostly clear, nevertheless, I have some concerns about the interpretation of the results and their real significance, since the conclusions are based on very subtle differences, where the molecular and biological significance remains to be proven. Therefore, I don't support the publication of this manuscript at this stage.

Major points:

1) The interaction with the multisynthetase complex is not surprising since tRNAs are 5'-Pme modified.

- ⇒ When we first obtained the ChemRAP data, we also made the connection between BCDIN3D methylating tRNA^{His} and the 5'-Pme reader being the multisynthetase complex. We tried very hard to establish this connection, but we have not been able to detect an interaction of EPRS with tRNA^{His} in cells. An explanation for a non tRNA^{His} target of BCDIN3D being bound to EPRS started to take shape when we established that it is the linker domain of EPRS, which does not bind tRNA, that binds the 5'-Pme RNA. Following the data in an unbiased manner, we were able to hypothesize that 5'-Pme mRNAs are bound by EPRS to regulate their translation.

Nevertheless, the authors could only confirm *in vitro* a slight preference for EPRS to bind 5'-Pme. In addition, from the components of the multisynthetase complex that bind to the two RNA baits in Fig 2, the authors excluded AIMP1, DARS, LARS and KARS from the discussion, which prefer to bind to 5'-P, and AIMP2, QARS, RARS, which don't bind at all. All these proteins were among the highest enriched proteins in the ChemRAP experiment. Therefore, considering the really slight preference of EPRS for the 5'-Pme bait, the *in vitro* pull-down (Fig 2C) doesn't validate the ChemRAP (Fig 2B).

- ⇒ We have included a detailed explanation about direct binders and passenger binders in the text and have also updated Fig 1A to clarify this point. In Fig 2C, which was designed to detect even weak RNA binding, only EPRS showed binding to the probe similar to what was observed with cellular extracts in the ChemRAP experiment. We went on to validate that 5'-Pme directly enhances RNA binding to EPRS through additional *in vitro* experiments, including UV crosslinking experiments in Fig 2D-E-F. Thus, altogether, the experiments in 2C-D-E-F validate the ChemRAP experiments.

In addition, the sentence in row 118: "EPRS binds directly to the 5'-Pme modification on the RNA, while ..." should be rephrased, since the authors observe only a slight preference of EPRS for 5'-Pme, but EPRS binds both to the 5'-Pme and 5'-P baits.

- ⇒ We have now made the requested text change in lines 133-134: "These results suggested that EPRS binding to RNA is enhanced by 5'-Pme, while MARS binding is not."

A competitive binding assay to show the preference of EPRS for 5'-Pme over 5'-P is missing here.

- ⇒ Performing these competitive binding assays requires large amounts of modified RNAs that we can no longer make because the German company IBA Lifesciences GmbH, which produced a wide range of custom modified RNA of premier quality, has now shut down their nucleic acid production to reduce costs. Nevertheless, our ChemRAP experiments in Figs 1-2A and *in vitro* binding experiments in Figs 2C-D-E-F, which include UV crosslinking assays, clearly show that EPRS binding to RNA is enhanced by 5'-Pme.

2) Row 137: It is not clear how the interaction between BCDIN3Df, MARS and EPRS could "independently support" the ChemRAP findings, since all the other multisynthetase complex members identified in the ChemRAP experiment are not tested (or not shown) here.

⇒ We have now included the requested results in the new Fig EV3.

3) Fig 4A. The authors write: "This image clearly shows that EPRS, which migrates at 187 ~170 KDa, interacts with RNA. Moreover, BCDIN3D-KO severely decreases the levels of RNAs directly interacting with EPRS." Here a quantification of RNA binding in the two genotypes it is needed.

⇒ We have now included the requested quantification in the updated Fig 3A.

4) The fact that α EPRS iCLIP-seq could not detect or identify changes in the known and much more abundant BCDIN3D RNA targets, like tRNAs and miRNAs, raises doubts on the specificity of the assay. Therefore, a validation of the results is needed. A possibility is that tRNAs and miRNAs could not be efficiently sequenced. α EPRS iCLIP-Northerns with probes specific to cognate tRNAs or known miRNAs could help.

⇒ We have now included the requested northern blot in the new Fig 3G. This northern blot, which was performed under co-IP conditions that preserve binding of tRNA^{Glu} and tRNA^{Met} to both EPRS and MARS, does not detect any binding to tRNA^{His}.

5) Figure 4D-E: Again, a validation with known BCDIN3D targets is needed.

⇒ We have now strengthened the corresponding results showing that BCDIN3D methylates the LRPPRC mRNA by performing a dual treatment of RNA with \pm Tobacco acid pyrophosphatase (TAP) and \pm Terminator. TAP decaps mRNA to 5'-P and Terminator degrades RNAs with 5'-P ends, but not 5'-Pme ends. We show that 20 to 25% of the LRPPRC mRNA is resistant to this dual treatment in a BCDIN3D-dependent manner, strengthening our results that LRPPRC is 5'-Pme by BCDIN3D in cells. These new results are shown in the new Fig 4E-F-G and described in lines 263-276.

6) Row 269: To show that the 5'UTR of LRPPRC mRNA is sufficient for "more translation" of the GFP reporter in BCDIN3D-KO cells, polysome profiling experiments are necessary, since the approach with GFP reporters used by the authors only shows that there is more protein, and not that the mRNA is translated more efficiently.

⇒ To address this key point raised by the reviewer, we have performed the requested analysis by polysome profiling that shows increased presence in the denser polysome fractions of LRPPRC mRNA in BCDIN3D-KO compared to control cells (new Fig 5C and lines 292-305). Additionally, we have performed ribosome footprinting coupled to next generation sequencing (Ribo-seq) in the presence of Harringtonine to specifically probe translation initiation. This Ribo-seq data shows increased initiation of translation of the LRPPRC mRNA in BCDIN3D-KO compared to control cells (new Fig 5D and lines 292-305).

7) Figure 4K-L and row 300: The hypothesis that the translation of LRPPRC requires the localization of its mRNA near mitochondria for proper mitochondrial transport is intriguing, but the evidence is based on very slight differences, that would need further validation in an independent cell line, or best a relevant more physiological model system (rather than cancer cells).

⇒ We agree that it would be ideal to show the relevance of this mechanism in a physiological system as we believe that our findings are relevant to the association of the BCDIN3D locus in humans with obesity and type II diabetes. It is our goal in a future publications to report our findings for the role of BCDIN3D in these processes. We have included these points in the conclusion.

In addition, clear prove, like the localization of the 5'-Pme modified fraction of LRPPRC mRNA next to the mitochondria or other membranes, is missing here. Moreover, if 5'-Pme acts as localization or translation signal of the RNA near mitochondria, how can this function be explained for the other known 5'-Pme modified RNAs, like tRNA and miRNA. To my knowledge, no enrichment of tRNA^{His} near mitochondria has been reported.

⇒ This question is very important but impossible for us to address given the current technical limitations of the RNA modification field and of the 5'-Pme sub-field. We are in the process of developing antibodies against 5'-Pme that could allow us to answer that question.

Minor points:

1) In Fig1b, H and L are swapped respect to Fig 1A. This is misleading, please clarify.

⇒ We have now made the swap to make **Fig 1A and 1B** consistent.

2) Figure 4 is really crowded. Better organization or splitting in two is needed.

⇒ We have now split the overcrowded Fig 4 in **two new Figs 4 and 5** as suggested.

3) In row 136 the authors write: "BCDIN3Df interacts in cells with MARS and EPRS and other members of the multisynthetase complex (Fig 3A and data not shown)." I think these data should be shown or not mentioned here.

⇒ We have now included the requested results in the **new Fig EV3**.

REFEREE #3:

Review of Ipas et al.: "ChemRAP uncovers specific mRNA translation regulation by the multisynthetase complex via RNA 5' phospho-methylation"

This study uses RNA pull-down and Mass Spectroscopy to identify protein binders to 5' methylated monophosphorylated RNA (5'Pme-RNA). The multi synthetase complex (MSC) is identified and the interaction with the complex is found to depend on the EPRS subunit. Furthermore, evidence is presented in favour of EPRS binding to specific mRNAs in a 5'Pme/BCDIN3D dependent manner with the LRPPRC mRNA-EPRS interaction being characterised further. These are interesting findings and the possibility that 5'Pme should regulate translation of specific mRNAs via recruitment of MSC could be an important and surprising discovery. The manuscript demonstrates that EPRS interacts with 5'Pme and this is the best characterised and most important finding of the study. Less convincing is the part of the manuscript showing that EPRS interact with specific messenger RNAs depending on interaction with the 5'Pme writer BCDIN3D. However, the presented data does not fully support the claims of the manuscript and additional experimentation will be needed to ensure that the conclusions are valid. In the current form, I therefore cannot recommend the publication this manuscript.

Major points:

1. SILAC MS experiment replicates. The authors state that they have three biological replicates of the experiments presented in Fig. 1C and 1D in main text (L78). These should be used for statistical analysis of the data and used for showing volcano plots or similar.

⇒ As suggested by the reviewer, we have now used these replicate data in statistical analyzes shown in the main Fig 2. The **updated Fig 2B now** shows the p value and the average enrichment fold change alongside the heatmap, as well as in a volcano plot.

2. The authors use CLIP-seq to identify the EPRS binding mRNAs but very few RNAs are detected, a positive control is missing and in combination this makes it difficult to interpret and trust the data. The EPRS linker has previously been shown to bind to RNA/the GAIT element (PMID 18374644, 22386318, 19647514 and more), these papers are not cited and discussed, and the GAIT elements are not detected in the analysis. The authors focus on the LRPPRC mRNA, which is one of the detected RNAs but based on the above the evidence for the interaction is not convincing. Phosphorylation of the EPRS linker may release EPRS from the MSC, allowing RNA interaction (with GAIT) and this seems to be of potential importance for the findings in this study, but is not analysed or discussed.

⇒ The positive control of the iCLIP-seq experiments with the anti-EPRS antibody is tRNA^{Glu}, which is a tRNA charged by EPRS.

⇒ We apologize for not making clear in the first version of the manuscript that one of the non-canonical functions of EPRS involving its linker domain is the GAIT system. We have now included an expanded explanation of the GAIT system in the text of the manuscript in **lines 144-153**, immediately following the description of our finding that EPRS binds 5'Pme through its linker domain. We also show our LC-MS/MS data that EPRSf does not interact with GAIT complex components in control or BCDIN3D-KO cells in the **new Fig 3E**. This shows that the disruption of EPRS binding to mRNAs in BCDIN3D-KO cells is not due to this pathway.

3. The study shows that BCDIN3D can 5'Pme modify a truncated version of the LRPPRC mRNA, which has an artificial dsRNA "blunt" end. However, in vivo the 3' end would extend further and in addition mRNAs are generally believed to be efficiently m7GpppN capped co-transcriptionally, meaning that a 5'Pme dependent mechanism of EPRS recruitment is hard to reconcile with normal mRNA processing. The methylation could occur after mRNA decapping but this remains speculative and clearly a demonstration that the LRPPRC mRNA (or any other mRNA) is 5'Pme modified to a biologically relevant level would strengthen the study a lot.

⇒ The reviewer brings up a very important question that we will strive to address in future work. We have now added an entire paragraph regarding this point in **lines 40-48**. As suspected by reviewer

#3, we do believe that the LRPPRC mRNA is modified with 5'-Pme after decapping, because LRPPRC is an mRNA that is heavily enriched in the P-bodies, where decapping and storage of mRNAs occurs (Hubstenberger et al., Molecular Cell 2017, PMID: 28965817). This is consistent with our new data in new Fig 4C showing that a very small portion of the LRPPRC mRNA is translated in cells.

- ⇒ We have now strengthened our results showing that BCDIN3D methylates the LRPPRC mRNA by performing a dual treatment of RNA with \pm Tobacco acid pyrophosphatase (TAP) and \pm Terminator. TAP decaps mRNA to 5'-P and Terminator degrades RNAs with 5'-P ends, but not 5'-Pme ends. We show that 20 to 25% of the LRPPRC mRNA is resistant to this dual treatment in a BCDIN3D-dependent manner, strengthening our results that LRPPRC is 5'-Pme by BCDIN3D in cells. These new results are shown in the new Fig 4E-F-G and described in lines 263-276. We have also extended our results by including the PGK1 mRNA in all of our analyses.

4. For the data in figure 3C, it would make sense to also show the non-MSF interactor to give some idea of the variation in the experiment.

- ⇒ As suggested by the reviewer, we have now included our LC-MS/MS data that EPRSf does not interact with GAIT complex components in control or BCDIN3D-KO cells in the new Fig 3E.

5. I appreciate that the authors mostly refrain from overstating their result and use "suggest", "may" and "consistent with", however I still think that both the title, abstract and discussion make claims that is not supported by the data.

- ⇒ To answer this comment, and as can be seen in our extensive revision and answers to the reviewers, we have significantly strengthened our data throughout the manuscript.

Date: 2nd Oct 23 12:16:11
Last Sent: 2nd Oct 23 12:16:11
Triggered By: Kelly Anderson
From: k.anderson@embojournal.org
To: b.xhemalce@austin.utexas.edu
Subject: Manuscript EMBOJ-2023-113533R-Q - Decision
Message: Dear Dr. Xhemalce,

Thank you for submitting your manuscript for re-consideration. I've now had a chance to confer with two referees and discuss with the editorial team and we've unfortunately concluded not to move forward with the publication of your manuscript.

While I appreciate one referee appears satisfied with the revised version, referee 1 refused to have a look at the new version and referee 3 remains unconvinced by the revision. Given that the initial reports were fairly negative, I do not find compelling reason to overrule the remaining concerns raised by referee 3. As this is the second rejection I must inform you this is the final decision on this manuscript.

Thank you in any case for the opportunity to consider this manuscript. I am sorry we cannot be more positive on this occasion, but we hope nevertheless that you will find our referees' comments helpful.

Yours sincerely,

Kelly M Anderson, PhD
Editor, The EMBO Journal
k.anderson@embojournal.org

Referee #2:

The manuscript has improved significantly in various ways and the authors have responded adequately to my concerns. I do not have additional comments.

Referee #3:

Review of Ipas et al. revision

In the revised version of the manuscript, authors have included additional data to strengthen their conclusions. However, the interpretation of the iCLIP data remains questionable and the added TAP+Terminator experiments does not convincingly demonstrate 5'Pme modification of LRPPRC mRNA or explain how this interaction would take place for a untruncated LRPPRC mRNA which would not have the secondary structure at the 5' end. Overall, the evidence for BCDIN3D regulating the interaction between EPRS and specific mRNAs is suggestive at best. Therefore, I still find that this manuscript requires additional experimental evidence or much more careful and conservative interpretation of the existing data before publication.

** As a service to authors, EMBO Press provides authors with the possibility to transfer a manuscript that one journal cannot offer to publish to another EMBO publication or the open access journal Life Science Alliance launched in partnership between EMBO Press, Rockefeller University Press and Cold Spring Harbor Laboratory Press. The full manuscript and if applicable, reviewers' reports, are automatically sent to the receiving journal to allow for fast handling and a prompt decision on your manuscript. For more details of this service, and to transfer your manuscript please click on Link Unavailable **

ANSWERS TO REFEREES

We would like to thank referees #2 and #3 for reviewing our revised manuscript. Our point-by-point responses are included below, with the original comments in bold, and our answers in regular text. We have also highlighted the changes in the main text of the manuscript in teal color.

Referee #2:

The manuscript has improved significantly in various ways and the authors have responded adequately to my concerns. I do not have additional comments.

⇒ We thank referee #2 for their positive review of our revised manuscript and their support for its publication.

Referee #3:

Review of Ipas et al. revision

In the revised version of the manuscript, authors have included additional data to strengthen their conclusions. However, the interpretation of the iCLIP data remains questionable and the added TAP+Terminator experiments does not convincingly demonstrate 5'Pme modification of LRPPRC mRNA or explain how this interaction would take place for a untruncated LRPPRC mRNA which would not have the secondary structure at the 5' end. Overall, the evidence for BCDIN3D regulating the interaction between EPRS and specific mRNAs is suggestive at best. Therefore, I still find that this manuscript requires additional experimental evidence or much more careful and conservative interpretation of the existing data before publication.

- ⇒ We first would like to thank referee #3 for acknowledging that our revision strengthened our conclusions.
- ⇒ We would also like to acknowledge that our conclusions could be strengthened by additional experiments.

1) *Interpretation of the iCLIP data:*

We showed by both iCLIP-seq and X-RIP that LRPPRC does not bind to EPRS in BCDIN3D-KO cells. Showing whether this is due to EPRS binding to the 5'-Pme LRPPRC mRNA would require us to:

- produce the crystal structure of the entire linker domain in complex with the 5'-Pme LRPPRC mRNA
- identify point mutation(s) that specifically disrupt(s) the binding of EPRS to 5'-Pme
- introduce the identified EPRS mutation in the EPRS genomic locus
- redo iCLIP-seq and all other key experiments in the paper

We are planning to perform these experiments, but they are not feasible given the timeframe and the difficulty of these experiments. We have now acknowledged this in our discussion. The addition of the corresponding text (see below) provides a much more careful and conservative interpretation of the existing data before publication, as requested by the reviewer. We have also changed the title of the paper and the last sentence of the abstract by removing the mention of EPRS and MSC.

- The new title of the paper is:
“ChemRAP uncovers specific mRNA translation regulation via RNA 5' phospho-methylation”.
- The added limitation in our discussion is:
“Additionally, while we showed by both iCLIP-seq and X-RIP that LRPPRC mRNA does not bind to EPRS in BCDIN3D-KO cells, we did not formally show whether the lack of binding of LRPPRC mRNA in BCDIN3D-KO cells is due to EPRS requiring 5'-Pme on the LRPPRC mRNA for efficient binding. Future work, based on co-crystal structure of EPRS linker domain with the 5'-Pme LRPPRC mRNA or *in silico* simulations, will need to identify point mutation(s) that specifically disrupt(s) the binding of EPRS linker domain to 5'-Pme. After introducing the identified EPRS mutation in the EPRS genomic locus, future work will also need to determine if the identified mutation(s) recapitulate the BCDIN3D-KO phenotype.”

2) *The added TAP+Terminator experiments does not convincingly demonstrate 5'Pme modification of LRPPRC mRNA*

We could not show that LRPPRC mRNA is 5'-Pme by mass spectrometry due to technical limitations of the field of RNA mass spectrometry. We remediated this limitation by showing that 20 to 25% of LRPPRC mRNA is resistant to dual TAP+Terminator treatment in a BCDIN3D-dependent manner and that BCDIN3D can methylate the 5' end of the LRPPRC mRNA *in vitro*. We would like to point out that even m⁷G capping of specific RNAs cannot be studied by mass spectrometry and is instead studied through indirect methods, such as enrichment with m⁷G antibody in a TAP-dependent manner.

3) *Explain how this interaction would take place for a untruncated LRPPRC mRNA which would not have the secondary structure at the 5' end.*

We believe that this note refers to the fact that we used a fragment of LRPPRC mRNA in our BCDIN3D RNA methyltransferase assays in Fig 4H:

H *in vitro* RNA methylation assay

We do not believe that the use of the LRPPRC mRNA fragment would be problematic, as the 3' end that follows the dsRNA portion is expected to point away from the BCDIN3D catalytic domain. Indeed, BCDIN3D has a paralog in humans, MePCE, that phospho-methylates the 5'-PPP of the 7SK snRNA. As can be seen in a representation of MePCE structure with the SL1p RNA derived from 7SK, the 3' end of the RNA point away from the structure of the MePCE S-Adenosyl Methionine binding domain (see below). Additionally, previous work with BCDIN3D and tRNA^{His} has shown that what is most important for BCDIN3D activity is that the first nucleotide is not base-paired and sticks out from the downstream dsRNA portion of tRNA^{His}.

(A)

(B)

(A) Structure of the MePCE S-Adenosyl Methionine binding domain complexed with 5'-PPP-7SL1p fragment of 7SK. (B) Schematic of SL1p fragment of 7SK used by Yang *et al.*, *Nature Chemical Biology*, 2018.

Dear Blerta,

Thank you for the transfer of your revised ms to EMBO reports. I have now heard back from both referees, and both in principle support the publication of your study now.

Both referees have a few more suggestions that I would like you to address, I paste their comments below and attach referee 3's comments in your point-by-point response to this email. Please let me know if you cannot or do not want to address these last suggestions.

In addition, a few editorial requests will also need to be addressed:

- Your ms has 5 main figures but is layed out as a full article. Please either add one more main figure, or combine the results and discussion sections to publish your study as a short report with a maximum of 29.000 characters. You can find more information about our article types in our guide to authors online.

- Please submit the ms as a word file without figures.

- Please add up to 5 keywords to the ms file.

- Please add a Data Availability Section (DAS) to the end of the Materials and Methods. If you have not deposited newly generated data in public databases, please mention this fact in the DAS.

- Please rename your conflict of interest statement to "Disclosure and Competing Interest Statement"

- This author is missing in our online ms submission system: Po-Chin Chiou. Please add.

- Please remove the author credits from the ms file. All credits need to be entered online during ms submission.

- The REFERENCE FORMAT is ok, but DOIs should only be used for preprints and datasets that have not been published yet, please correct.

- Please submit with your final ms files a completed author checklist that can be found here : <https://www.embopress.org/page/journal/14693178/authorguide>. The completed author checklist will also be part of the transparent peer-review process file (RPF).

- Please enter all funding info in our online submission system.

- Please upload all main and EV figures individually, as one file per figure.

- A callout for Fig 5l is missing, please add.

- The 3 suppl. tables need to be renamed to Dataset EV1-EV3; their legends need to be provided in the Excel files as a separate tab/sheet, and the callouts need to be corrected in the ms text.

- The suppl. file with Materials and Methods, some tables and 9 EV figures needs to be called Appendix, and needs a table of content with page numbers and the nomenclature should be Appendix Table S1, etc. Appendix Figure S1, etc. (for the figures that won't be EV figures); the callouts in the ms also need to be updated accordingly. In addition to the Appendix you can chose 5 EV figures that are embedded in the online ms file. You can find more information in our guide to authors online.

- The main materials and methods need to be moved to the main ms file.

- The manuscript sections should be in the following order: Title page - Abstract & Keywords - Introduction - Results - Discussion - Materials & Methods - Data Availability - Acknowledgments - Disclosure Statement & Competing Interests - References - Figure Legends - Tables with legends - Expanded View Figure Legends.

- It seems that the western blots in Fig 5 and EV5 are re-used, but this is not cited in the figure legends. Please clarify and correct.

- Please address the following figure legend issues:

1. Please note that information related to n is missing in the legend of figure 4h.

2. Please note that n=2 in figures 3c, e; 5c; EV 5. IF n=2 no statistics can be calculated, but all individual datapoints can be

shown, along with their mean.

3. Although 'n' is provided, please describe the nature of entity for 'n' in the legends of figures 3e; 4d, g; 5c, h; EV 5.
4. Please note that the error bar is not defined in the legend of figure 4h.
5. Please define the annotated p values **/* in the legend of figures 4d, g; EV 8; EV 9; as appropriate.
6. Please indicate the statistical test used for data analysis in the legend of figure 3c.
7. Please note that for the figure 5b, p-value and statistical test is indicated in the legend. However, comparison for the same, "" has not been represented in the figure. Please rectify this in the figure or legend as applicable.
8. Please note that scale bar and its definition are missing for figure 5i.
9. Please note that the asterisk "" is not defined in the legend of figures 4h; 5c; EV 8. This needs to be rectified.

EMBO press papers are accompanied online by A) a short (1-2 sentences) summary of the findings and their significance, B) 2-3 bullet points highlighting key results and C) a synopsis image that is exactly 550 pixels wide and 200-600 pixels high (the height is variable). You can either show a model or key data in the synopsis image. Please note that text needs to be readable at the final size. Please send us this information along with the final manuscript.

I look forward to seeing a new revised version of your manuscript as soon as possible.

Comments from referee 2:

I was already happy with the revised and significantly improved manuscript in the last round of review at The EMBO Journal. Thus, I fully support publication of the study at EMBO reports.

Regarding the concern of Referee 3 you refer to, I think that the way how the authors interpret now the data and phrase it in the text provides a more cautious description of the results and addresses already appropriately the concern of the reviewer. However, even if the TAP+Terminator experiments are not fully addressing the problem, the results point in the direction described by the authors and to the 5'-Pme of the LRPPRC mRNA. A possible control experiment would be a Northern with a probe against a known methylated 5' tRNA, like t tRNAHis (5'-Pme) for the samples in Figure 4E-F, this will render the assay more convincing.

Comments from referee 3:

I have read the updated version and I find that the claims have been moderated sufficiently and that the paper is suitable for publication with very minor changes. I have attached a PDF with a brief reply to the author comments.

Review of Ipas et al. revision

In the revised version of the manuscript, authors have included additional data to strengthen their conclusions. However, the interpretation of the iCLIP data remains questionable and the added TAP+Terminator experiments does not convincingly demonstrate 5'Pme modification of LRPPRC mRNA or explain how this interaction would take place for a untruncated LRPPRC mRNA which would not have the secondary structure at the 5' end. Overall, the evidence for BCDIN3D regulating the interaction between EPRS and specific mRNAs is suggestive at best. Therefore, I still find that this manuscript requires additional experimental evidence or much more careful and conservative interpretation of the existing data before publication.

⇒ We first would like to thank referee #3 for acknowledging that our revision strengthened our conclusions.

⇒ We would also like to acknowledge that our conclusions could be strengthened by additional experiments.

REFEREE: With the more conservative interpretation of results, I find that this paper improved a lot and with minor changes is suitable for publication.

1) *Interpretation of the iCLIP data:*

We showed by both iCLIP-seq and X-RIP that LRPPRC does not bind to EPRS in BCDIN3D-KO cells. Showing whether this is due to EPRS binding to the 5'-Pme LRPPRC mRNA would require us to:

- produce the crystal structure of the entire linker domain in complex with the 5'-Pme LRPPRC mRNA
- identify point mutation(s) that specifically disrupt(s) the binding of EPRS to 5'-Pme
- introduce the identified EPRS mutation in the EPRS genomic locus
- redo iCLIP-seq and all other key experiments in the paper

We are planning to perform these experiments, but they are not feasible given the timeframe and the difficulty of these experiments. We have now acknowledged this in our discussion. The addition of the corresponding text (see below) provides a much more careful and conservative interpretation of the existing data before publication, as requested by the reviewer. We have also changed the title of the paper and the last sentence of the abstract by removing the mention of EPRS and MSC.

REFEREE: I appreciate the more conservative interpretation of results and discussion and agree that the experiments required to demonstrate that 5'Pme is linked to the EPRS binding are time consuming.

- The new title of the paper is:
“ChemRAP uncovers specific mRNA translation regulation via RNA 5' phospho-methylation”.
- The added limitation in our discussion is:
“Additionally, while we showed by both iCLIP-seq and X-RIP that LRPPRC mRNA does not bind to EPRS in BCDIN3D-KO cells, we did not formally show whether the lack of binding of LRPPRC mRNA in BCDIN3D-KO cells is due to EPRS requiring 5'-Pme on the LRPPRC mRNA for efficient binding. Future work, based on co-crystal structure of EPRS linker domain with the 5'-Pme LRPPRC mRNA or *in silico* simulations, will need to identify point mutation(s) that specifically disrupt(s) the binding of EPRS linker domain to 5'-Pme. After introducing the identified EPRS mutation in the EPRS genomic locus, future work will also need to determine if the identified mutation(s) recapitulate the BCDIN3D-KO phenotype.”

2) *The added TAP+Terminator experiments does not convincingly demonstrate 5'Pme modification of LRPPRC mRNA*

We could not show that LRPPRC mRNA is 5'-Pme by mass spectrometry due to technical limitations of the field of RNA mass spectrometry. We remediated this limitation by showing that 20 to 25% of

LRPPRC mRNA is resistant to dual TAP+Terminator treatment in a BCDIN3D-dependent manner and that BCDIN3D can methylate the 5' end of the LRPPRC mRNA *in vitro*. We would like to point out that even m⁷G capping of specific RNAs cannot be studied by mass spectrometry and is instead studied through indirect methods, such as enrichment with m⁷G antibody in a TAP-dependent manner.

REFEREE: These results are shown in Fig.4G. From the method section it is somewhat unclear to me which detection method was used. I guess that the plot is summarising northern blots, but how this is normalised and quantified is unclear. The plot does show quite some variability and I think it is central for the conclusion that the mRNA is 5' phosphor-methylated in cells. I would recommend adding the northern blots to the supplementary materials, thereby allowing readers to judge the data themselves and also update the methods section with the details about the detection and normalisation.

3) ***Explain how this interaction would take place for a untruncated LRPPRC mRNA which would not have the secondary structure at the 5' end.***

We believe that this note refers to the fact that we used a fragment of LRPPRC mRNA in our BCDIN3D RNA methyltransferase assays in Fig 4H:

H *in vitro* RNA methylation assay

We do not believe that the use of the LRPPRC mRNA fragment would be problematic, as the 3' end that follows the dsRNA portion is expected to point away from the BCDIN3D catalytic domain. Indeed, BCDIN3D has a paralog in humans, MePCE, that phospho-methylates the 5'-PPP of the 7SK snRNA. As can be seen in a representation of MePCE structure with the SL1p RNA derived from 7SK, the 3' end of the RNA point away from the structure of the MePCE S-Adenosyl Methionine binding domain (see below). Additionally, previous work with BCDIN3D and tRNA^{His} has shown that what is most important for BCDIN3D activity is that the first nucleotide is not base-paired and sticks out from the downstream dsRNA portion of tRNA^{His}.

(A)

(B)

(A) Structure of the MePCE S-Adenosyl Methionine binding domain complexed with 5'-PPP-7SL1p fragment of 7SK. (B) Schematic of SL1p fragment of 7SK used by Yang *et al.*, *Nature Chemical Biology*, 2018.

REFEREE: Makes sense.

SUBMISSION CHECKLIST

Dear Blerta,

Thank you for the transfer of your revised ms to EMBO reports. I have now heard back from both referees, and both in principle support the publication of your study now.

⇒ Thank you for the good news that both referees support the publication of our work. Please see below our answers to both referees' and editors' comments.

Both referees have a few more suggestions that I would like you to address, I paste their comments below and attach referee 3's comments in your point-by-point response to this email. Please let me know if you cannot or do not want to address these last suggestions.

⇒ To answer the referees' final minor comments concerning our data in former Fig 4G, we have now provided 3 independent biological replicates of the TAP+Terminator experiments in the new Fig 5C, added explanations specifically on how analysis was done in Materials and Methods (section 18) and Fig 5C figure legend, and added all source data of our paper.

In addition, a few editorial requests will also need to be addressed:

⇒ Please see our answers to the editorial requests here:

- Your ms has 5 main figures but is layed out as a full article. Please either add one more main figure, or combine the results and discussion sections to publish your study as a short report with a maximum of 29.000 characters. You can find more information about our article types in our guide to authors online.

⇒ We have now a total of 6 Figures in the manuscript.

- Please submit the ms as a word file without figures.

⇒ We have submitted a word file of our manuscript without figures as requested.

- Please add up to 5 keywords to the ms file.

⇒ We have added 5 key words to the ms file as requested.

- Please add a Data Availability Section (DAS) to the end of the Materials and Methods. If you have not deposited newly generated data in public databases, please mention this fact in the DAS.

⇒ We have added this information as requested.

- Please rename your conflict of interest statement to "Disclosure and Competing Interest Statement"

⇒ We have renamed this statement as requested.

- This author is missing in our online ms submission system: Po-Chin Chiou. Please add.

⇒ We have now added Po-Chin Chiou in the online submission system as requested.

- Please remove the author credits from the ms file. All credits need to be entered online during ms submission.

⇒ We have now removed the author credits from the ms file.

- The REFERENCE FORMAT is ok, but DOIs should only be used for preprints and datasets that have not been published yet, please correct.

⇒ We have now removed the DOIs from the references.

- Please submit with your final ms files a completed author checklist that can be found here : <<https://www.embopress.org/page/journal/14693178/authorguide>>; The completed author checklist will also be part of the transparent peer-review process file (RPF).

⇒ We have submitted the author checklist.

- Please enter all funding info in our online submission system.

⇒ We have now added all the funding in the online submission system.

- Please upload all main and EV figures individually, as one file per figure.

⇒ We have now uploaded all main figures individually.

- A callout for Fig 5l is missing, please add.

⇒ We have now added the missing callout for former Fig 5l (now Fig 6l).

- The 3 suppl. tables need to be renamed to Dataset EV1-EV3; their legends need to be provided in the Excel files as a separate tab/sheet, and the callouts need to be corrected in the ms text.

⇒ We have now renamed these dataset tables throughout the manuscript.

- The suppl. file with Materials and Methods, some tables and 9 EV figures needs to be called Appendix, and needs a table of content with page numbers and the nomenclature should be Appendix Table S1, etc. Appendix Figure S1, etc. (for the figures that won't be EV figures); the callouts in the ms also need to be updated accordingly. In addition to the Appendix you can chose 5 EV figures that are embedded in the online ms file. You can find more information in our guide to authors online.

⇒ We have now renamed the supplemental file Appendix and the Tables and Figures as Appendix Table S1 to S4, and Appendix Figure S1 to S9.

- The main materials and methods need to be moved to the main ms file.

⇒ We have now moved Material and Methods to the main manuscript file.

- The manuscript sections should be in the following order: Title page - Abstract & Keywords - Introduction - Results - Discussion - Materials & Methods - Data Availability - Acknowledgments - Disclosure Statement & Competing Interests - References - Figure Legends - Tables with legends - Expanded View Figure Legends.

⇒ We have followed this order in the manuscript.

- It seems that the western blots in Fig 5 and EV5 are re-used, but this is not cited in the figure legends. Please clarify and correct.

⇒ We have now referred to the data in Fig EV5 (now Appendix Fig S5) in the legend of Fig 5C (now Fig 6C).

- Please address the following figure legend issues:

1. Please note that information related to n is missing in the legend of figure 4h.

⇒ We have now clearly indicated the n in Fig 4H (now Fig 5D).

2. Please note that n=2 in figures 3c, e; 5c; EV 5. IF n=2 no statistics can be calculated, but all individual datapoints can be shown, along with their mean.

⇒ We have now modified the graphs to conform to this policy.

3. Although 'n' is provided, please describe the nature of entity for 'n' in the legends of figures 3e; 4d, g; 5c, h; EV 5.

⇒ We have now defined the nature of entity for 'n' in the legends of Fig 3E, 4D, 4G (now 5C), 5C (now 6C), 5H (now 6H), EV5 (now Appendix Fig S5).

4. Please note that the error bar is not defined in the legend of figure 4h.

⇒ We have now defined the nature of error bars (i.e. SD) in the legend of Fig 4H (now 5D).

5. Please define the annotated p values **/* in the legend of figures 4d, g; EV 8; EV 9; as appropriate.

⇒ We have now the annotated p values **/* in the legends of 4D, 4G (now 5C), EV8 (now Appendix Fig S8), EV9 (now Appendix Fig S9).

6. Please indicate the statistical test used for data analysis in the legend of figure 3c.

⇒ This is no longer indicated since we had to remove the statistical analysis (see point 2).

7. Please note that for the figure 5b, p-value and statistical test is indicated in the legend. However, comparison for the same, "" has not been represented in the figure. Please rectify this in the figure or legend as applicable.

⇒ We have now places the missing "*" on Fig 5B (now Fig 6B).

8. Please note that scale bar and its definition are missing for figure 5i.

⇒ We have now placed the scale bar on the microscopic images on Fig 5I (now 6I).

9. Please note that the asterisk "" is not defined in the legend of figures 4h; 5c; EV 8. This needs to be rectified.

⇒ We have now removed the asterisks on Fig 4H (now 5D) as we had more space to properly mark the migration of RNA#1 and #2, and we have defined the asterisks on Fig 5C (now Fig 6C) and EV8 (now Appendix Fig S8).

EMBO press papers are accompanied online by A) a short (1-2 sentences) summary of the findings and their significance, B) 2-3 bullet points highlighting key results and C) a synopsis image that is exactly 550 pixels wide and 200-600 pixels high (the height is variable). You can either show a model or key data in the synopsis image. Please note that text needs to be readable at the final size. Please send us this information along with the final manuscript.

⇒ We have now provided a 2-sentence summary of our findings, 3 bullet points, and a 550x550 pixels synopsis image.

I look forward to seeing a new revised version of your manuscript as soon as possible.

**Best regards,
Esther**

**Esther Schnapp, PhD
Senior Editor
EMBO reports**

Comments from referee 2:

I was already happy with the revised and significantly improved manuscript in the last round of review at The EMBO Journal. Thus, I fully support publication of the study at EMBO reports.

Regarding the concern of Referee 3 you refer to, I think that the way how the authors interpret now the data and phrase it in the text provides a more cautious description of the results and addresses already appropriately the concern of the reviewer. However, even if the TAP+Terminator experiments are not fully addressing the problem, the results point in the direction described by the authors and to the 5'-Pme of the LRPPRC mRNA. A possible control experiment would be a Northern with a probe against a known methylated 5' tRNA, like tRNA^{His} (5'-Pme) for the samples in Figure 4E-F, this will render the assay more convincing.

⇒ We thank referee #2 for supporting publication of our work. As mentioned above, to strengthen our data in former Fig 4G, we have now provided 3 independent biological replicates of the TAP+Terminator experiments in the new Fig 5C, added explanations specifically on how analysis was done in Material and Methods and Fig 5C figure legend, and added all source data of our paper. Given that treatment with Terminator doesn't digest mature tRNA, regardless of their 5' end status (Reinsborough *et al.*, PLOS Genetics 2019 and Devanathan *et al.*, 2021), we unfortunately cannot utilize northern blotting with tRNA^{His} as a control for the Terminator experiments. We have instead used tRNA resistance to Terminator treatment as a positive control for RNA recovery after enzymatic treatment of RNA.

Comments from referee 3:

I have read the updated version and I find that the claims have been moderated sufficiently and that the paper is suitable for publication with very minor changes. I have attached a PDF with a brief reply to the author comments.

(From PDF) With the more conservative interpretation of results, I find that this paper improved a lot and with minor changes is suitable for publication.

⇒ We thank referee #3 for supporting publication of our work in principle. We have answered their final minor comment here below.

1) (From PDF) I appreciate the more conservative interpretation of results and discussion and agree that the experiments required to demonstrate that 5'Pme is linked to the EPRS binding are time consuming.

⇒ We thank referee #3 for suggesting a more conservative interpretation of our results, which increases the rigor of our paper.

2) (From PDF) These results are shown in Fig.4G. From the method section it is somewhat unclear to me which detection method was used. I guess that the plot is summarising northern blots, but how this is normalised and quantified is unclear. The plot does show quite some variability and I think it is central for the conclusion that the mRNA is 5' phosphor-methylated in cells. I would recommend adding the northern blots to the supplementary materials, thereby allowing readers to judge the data themselves and also update the methods section with the details about the detection and normalisation.

⇒ As mentioned above, to strengthen our data in former Fig 4G, we have now provided 3 independent biological replicates of the TAP+Terminator experiments in the new Fig 5C, added explanations specifically on how analysis was done in Material and Methods and Fig 5C figure legend, and added all source data of our paper.

3) (From PDF) Makes sense.

⇒ We thank referee #3 for acknowledging the validity of our reasoning.

Dr. Blerta Xhemalce
University of Texas at Austin
Molecular Biosciences
2500 Speedway Stop A4800
Austin, TX 78712
United States

Dear Dr. Xhemalce,

I am very pleased to accept your manuscript for publication in the next available issue of EMBO reports. Thank you for your contribution to our journal.

Yours sincerely,
